# CD3-engaging bispecific antibodies trigger a paracrine regulated wave of T-cell recruitment for effective tumor killing
Chen-Yi Liao [1], Patrick Engelberts[2], Andreea Ioan-Facsinay[2], Janna Eleonora Klip[1], Thomas Schmidt [3], Rob Ruijtenbeek[2] & Erik H. J. Danen [1] ✉

The mechanism of action of bispecific antibodies (bsAbs) directing T-cell immunity to solid tumors is incompletely understood. Here, we screened a series of CD3xHER2 bsAbs using extracellular matrix (ECM) embedded breast cancer tumoroid arrays exposed to healthy donor-derived T-cells. An initial phase of random T-cell movement throughout the ECM (day 1–2), was followed by a bsAb-dependent phase of active T-cell recruitment to tumoroids (day 2–4), and tumoroid killing (day 4–6). Low affinity HER2 or CD3 arms were compensated for by increasing bsAb concentrations. Instead, a bsAb binding a membrane proximal HER2 epitope supported tumor killing whereas a bsAb binding a membrane distal epitope did not, despite similar affinities and intra-tumoroid localization of the bsAbs, and efficacy in 2D co-cultures. Initial T-cell-tumor contact through effective bsAbs triggered a wave of subsequent T-cell recruitment. This critical surge of T-cell recruitment was explained by paracrine signaling and preceded a full-scale T-cell tumor attack.

Breast cancer represents a heterogeneous disease with multiple clinically relevant subtypes originating from luminal or basal epithelial cells in the mammary ducts[1,2]. In over 20% of breast carcinomas, the *ERBB2* gene is amplified leading to overexpression of human epidermal growth factor receptor 2 (HER2), which is associated with poor prognosis[3,4]. Treatment with the HER2 antibody trastuzumab (Herceptin) has been developed for patients with metastatic breast cancer displaying HER2 overexpression, as determined by immunohistochemistry (IHC) or *ERBB2* amplification by fluorescence in situ hybridization (FISH)[5–9]. After the successful clinical use of trastuzumab, other HER2-targeting drugs have emerged including antibodies, kinase inhibitors, and antibody-drug conjugates. Importantly, approximately 50% of breast cancers lack *ERBB2* gene amplification but do express HER2, are classified as "HER2 low", and potentially also benefit from advanced therapies targeting HER2[10–12].

Although the mechanisms of action of therapeutic antibodies such as trastuzumab underlying clinical efficacy are incompletely understood, preclinical studies indicated that engagement of immune cells contributes to efficacy[13]. Bispecific antibodies (bsAbs) binding the CD3 T-cell receptor (TCR) subunit with one arm and a tumor associated antigen (TAA) with the other arm can recruit T-cells to tumor cells and trigger formation of an immune synapse with high similarity to that formed by natural TCR-major histocompatibility (MHC)/TAA interactions[14]. This provides a way to redirect immune cell cytotoxicity towards tumors irrespective of TCR specificity[15,16]. Many CD3xTAA bsAbs are undergoing clinical testing, including two CD3xHER2 bsAbs[16,17]. Thus far, T-cell engaging CD3xTAA bsAbs have shown clinical benefit in the context of non-Hodgkin's lymphoma and acute lymphoblastic leukemia[18,19].

Application of bsAbs to the treatment of solid tumors lags behind that of hematological tumors. Nevertheless, several T-cell-engaging bsAbs are being investigated in clinical trials in the context of solid tumors[16]. Low expression of TAAs at the surface of solid tumors, toxicity associated with systemic cytokine release or on-target off-tumor effects, expression of suppressive immune checkpoint molecules, and the immune suppressive solid tumor microenvironment may hinder efficacy[15,16,20]. Strategies to circumvent these hurdles are being developed. For example, preclinical work using orthotopic breast cancer models in mice has demonstrated that immune checkpoint inhibitors can improve the response to CD3xHER2 bsAbs[21]. Fine-tuning of bsAb affinities also appears critical: while sufficient affinities in the CD3 and TAA binding arm are important, it has been shown for a CD3xHER2 bsAb that too

[1]Leiden Academic Centre for Drug Research, Leiden University, Leiden, the Netherlands. [2]Genmab, Utrecht, the Netherlands. [3]Leiden Institute of Physics, Leiden University, Leiden, the Netherlands. ✉e-mail: e.danen@lacdr.leidenuniv.nl

high affinity for CD3 led to a concentration in secondary lymphatic tissues while precluding recruitment to the tumor[22].

BsAbs are typically evaluated using tumor-T-cell mixtures in suspension or co-cultures of a monolayer of tumor cells with T-cells. This experimental setup allows high-throughput drug testing but lacks biomechanical and biochemical features representative of the three-dimensional (3D) solid tumor architecture. The emergence of 3D models represents a significant advancement in this area[23–25]. Co-cultures of patient-derived tumor organoids with allogeneic or autologous lymphocytes allow for in vitro personalized drug testing but lack high throughput capacity, and have a level of complexity that limits their use in the initial screening of high number of bsAbs[26–30]. Mixed 3D co-cultures of tumor spheroids with T-cells can be produced with higher throughput for in vitro drug testing[31–33] but do not allow monitoring of recruitment of T-cells from the environment towards the tumoroid.

Here, we developed a screening platform of extracellular matrix (ECM) embedded tumoroid arrays that were exposed to human healthy donor-derived T-cells in the presence of bsAbs. Automated confocal microscopy and quantitative 3D image analysis provided high throughput quantitative data on T-cell recruitment to-, and subsequent killing of tumoroids over a time frame of approximately 1 week. We screened CD3xHER2 bsAbs displaying different affinities to CD3 or HER2, or binding to different epitopes on HER2. We find that besides affinity, the TAA epitope targeted by the bsAbs determines their ability to initiate a wave of active recruitment of T-cells from the surrounding environment, triggering a full-scale immune attack against a 3D tumor structure. Our work indicates that 2D co-cultures fail to capture the full complexity of bsAb-induced T-cell antitumor cytotoxicity.

## Results
### ECM embedded tumoroid arrays for quantitative analysis of T-cell recruitment and tumoroid killing

We employed automated image guided injection robotics for rapid and reproducible generation of ECM embedded tumoroids using our previously established technology[25]. Tumoroids were printed with an initial diameter of ~150 μm at identical x-y-z coordinates in 1.5 mg/mL collagen gel in 96-well plates (Fig. 1A). We first applied a concentration range of cisplatin to establish quantitative analysis of tumoroid viability (Supplementary Fig 1). ECM embedded tumoroids were generated from Hoechst labeled BT474 or MDA-MB-231 cells. Tumoroid cultures were exposed to 0, 5, 10, 20 μg/mL cisplatin. Propidium iodide (PI) was included to label dead cells. Making use of the identical position of the ECM embedded tumoroid in each well, automated confocal microscopy was used to image the tumoroids at 24 and 48 h post-treatment, creating stacks of images at 10 μm spacing along the entire z-axis of the tumoroid. We established an automated 3D image analysis pipeline to process and analyze the image datasets using Cell Profiler (Fig. 1B). In each plane, tumor nuclei were detected (Hoechst) and subsequently scored as PI positive or negative. Tumoroid viability was quantified by summarizing the data of all images in the stack to obtain the percentage of PI positive tumor cells within the entire tumoroid. For both cell lines, a concentration dependent increase in cell death was observed at 24 h and a steeper increase that maximized at 10 μg/mL was observed at 48 h (Supplementary Fig 1).

Having established automated imaging and quantitative image analysis, we extended this approach to the analysis of T-cell recruitment and T-cell mediated tumoroid killing (Fig. 1). ECM embedded tumoroids were generated from HER2 + BT474 tumor cells. Human healthy donor-derived T-cells were added on top of the collagen gels in absence or presence of a CD3xHER2 bsAb. T-cells were labeled with cell tracker CMFDA. This allowed for analysis of T-cell recruitment to the tumoroid and tumor viability at different timepoints (Fig. 1A). Confocal image stacks with 10 μm spacing along the entire z-axis of the tumoroid were captured. Each plane was analyzed in CellProfiler for PI positive tumor cells and presence of T-cells, to quantify cell death and the total count of T-cells recruited to the tumoroid (Fig. 1B).

### Localization of bsAbs to ECM-embedded HER2+ tumoroids depends on affinity of the HER2 arm

In preparation of screening a panel of Fc-inert CD3xHER2 IgG1 bsAbs that were generated using controlled Fab-arm exchange, termed DuoBody® technology[34–36] (see Table 1 for the mutations used), we first validated penetration of the bsAbs in the fibrillar collagen scaffolds and their concentration at HER2 + BT474 tumoroids. ECM embedded BT474

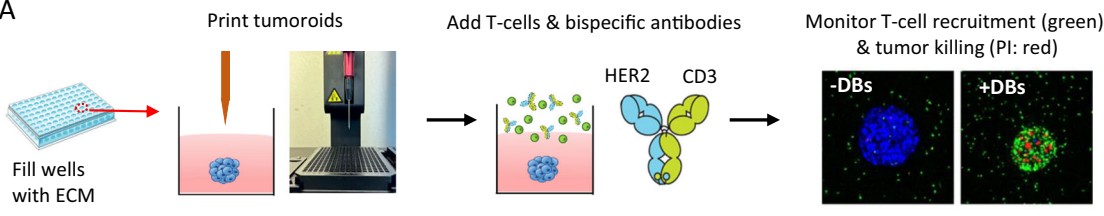

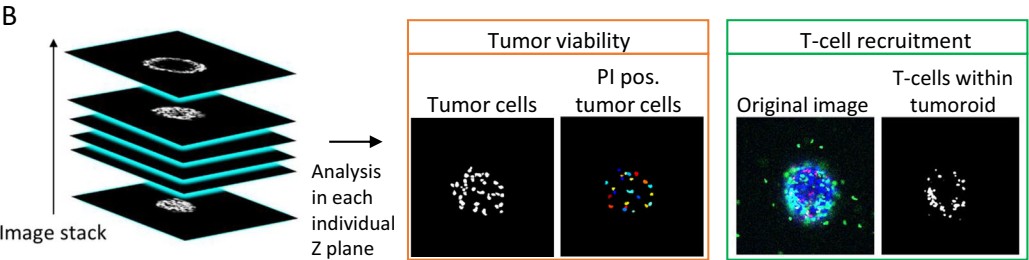

**Fig. 1 | High-throughput screening platform to evaluate bsAb-induced T-cell-mediated tumoroid killing. A** Workflow for bsAb screening: tumoroids are printed at identical x-y-z positions in 96-well plates prefilled with collagen gel, followed by the addition of T-cells and bsAbs on top of the gel. Migration and recruitment of T-cells (CMFDA; green) to tumoroids (Hoechst; blue) and tumor killing (PI; red) are monitored using automated confocal microscopy at different time points. The cartoon of the 96-well plate was obtained from Servier Medical Art (https://smart.servier.com/), which is licensed under a Creative Commons Attribution 4.0 Unported License (https://creativecommons.org/licenses/by/4.0/). **B** Schematic representation of the automated image analysis procedure: stacks of confocal images are captured at 10 μm vertical spacing throughout the z-axis of the tumoroid. Tumor nuclei and T-cells are identified separately and tumor nuclei with or without over-lapping PI signal are counted in each z-section. Tumor killing is determined by calculating the percentage of PI positive tumor cells in the entire tumoroid. T-cell recruitment is determined as the sum of all T-cells within the tumoroid counted in each z-section. PI Propidium iodide.

## Table 1 | Characteristics of bsAbs used in this study

| Description | Ab code | Clone CD3-arm | Clone HER2-arm | kD CD3 (nM)* | Apparent affinity EC50 HER2 (nM)** | kD HER2 (nM)*** |
|---|---|---|---|---|---|---|
| bsAb CD3$_{Wt}$ × HER2$_{Herceptin}$ | BisG1-huCACAO-FEAL/Herceptin-FEAR | huCACAO | Herceptin | 15 | 5.7 | 0.1 |
| bsAb CD3$_{Wt}$ × HER2$_{LbD2}$ | BisG1-huCACAO-FEAL/Herceptin-FEAR LbD2 | huCACAO | LbD2 | 15 | | 100 |
| bsAb CD3$_{Wt}$ × HER2$_{153}$ | BisG1-huCACAO-FEAL/153-FEAR | huCACAO | 153 | 15 | 4.2 | |
| bsAb CD3$_{Wt}$ × HER2$_{169}$ | BisG1-huCACAO-FEAL/169-FEAR | huCACAO | 169 | 15 | 6.1 | |
| bsAb CD3$_{Wt}$ × ctrl | BisG1-huCACAO-FEAL/b12-FEAR | huCACAO | b12 | 15 | N.A. | |
| bsAb CD3$_{Low}$ × HER2$_{Herceptin}$ | BisG1-huCACAO-H101G-FEAL/Herceptin-FEAR | huCACAO-H101G | Herceptin | 310 | 5.7 | 0.1 |
| bsAb CD3$_{Low}$ × HER2$_{LbD2}$ | BisG1-huCACAO-H101G-FEAL/Herceptin-FEAR LbD2 | huCACAO-H101G | LbD2 | 310 | | 100 |
| bsAb CD3$_{Low}$ × HER2$_{153}$ | BisG1-huCACAO-H101G-FEAL/153-FEAR | huCACAO-H101G | 153 | 310 | 4.2 | |
| bsAb CD3$_{Low}$ × HER2$_{169}$ | BisG1-huCACAO-H101G-FEAL/169-FEAR | huCACAO-H101G | 169 | 310 | 6.1 | |
| bsAb CD3$_{Low}$ × ctrl | BisG1-huCACAO-H101G-FEAL/b12-FEAR | huCACAO-H101G | b12 | 310 | N.A. | |
| bsAb ctrl × HER2$_{Herceptin}$ | BisG1-b12-FEAL/Herceptin-FEAR | b12 | Herceptin | N.A. | 5.7 | 0.1 |
| bsAb ctrl × HER2$_{LbD2}$ | BisG1-b12-FEAL/Herceptin-FEAR LbD2 | b12 | LbD2 | N.A. | | 100 |
| bsAb ctrl × HER2$_{153}$ | BisG1-b12-FEAL/153-FEAR | b12 | 153 | N.A. | 4.2 | |
| bsAb ctrl × HER2$_{169}$ | BisG1-b12-FEAL/169-FEAR | b12 | 169 | N.A. | 6.1 | |

*Wt* high affinity binding to CD3, *low* low affinity binding to CD3, *ctrl* non-binding arm using HIV-1 gp120-specific mAb IgG1-b12, *Herceptin* interaction with HER2 at Herceptin epitope, *LbD2* low affinity interaction with HER2 at Herceptin epitope due to light chain bD2 mutation, *153* high affinity interaction with membrane distal HER2 epitope "153", *169* high affinity interaction with membrane proximal HER2 epitope "169", *N.A.* not applicable, *FEAL* L234F, L235E, D265A Fc-inert with F405L mutations, *FEAR* L234F, L235E, D265A Fc-inert with K409R mutations.

*kD values for WT CD3 arm described in patent WO-2017/009442 and for H101G mutant in ref. 59.
**kD values for Herceptin and LbD2 mutant arms determined in ref. 60.
***Apparent affinities determined in house.

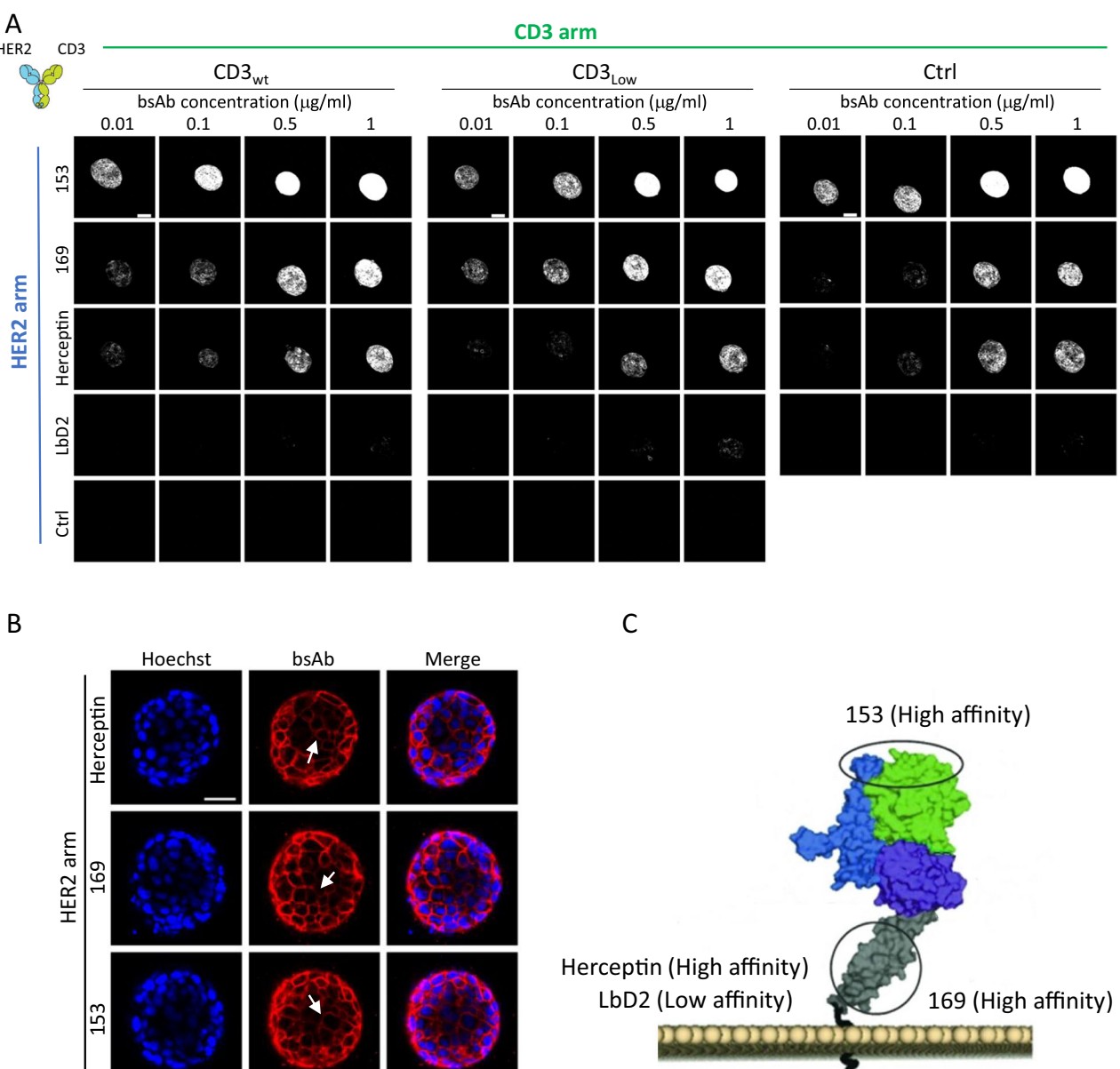

**Fig. 2 | Localization of bsAbs to tumoroids. A** A concentration range of Alexa Fluor 647-conjugated CD3xHER2 bsAbs varying in affinity and epitope recognition site on either arm, were added on top of collagen gels containing BT474 (HER2+) tumoroids. Localization of bsAbs to the tumoroids was analyzed after 24 h by confocal microscopy. Images show maximum projections. Bar = 100 μm. **B** Localization of three different bsAbs combining a CD3 high affinity arm with the indicated HER2 arms in the BT474 tumoroid. White arrows indicate penetration of bsAbs into the tumoroids. Blue, Hoechst; red, Alexa Fluor 647-conjugated bsAb. Images were obtained from a single z-section through the center of the tumoroid. Bar = 50 μm. **C** Cartoon showing the location on the HER2 receptor of the epitopes recognized by the different HER2 arms and whether the interactions are high or low affinity.

tumoroids were exposed to a concentration range (0.01, 0.1, 0.5, 1 μg/mL) of various Alexa Fluor 647-conjugated CD3xHER2 bsAbs that were added on top of the collagen gels. BsAb localization was assessed 24 h later using confocal microscopy. BsAbs binding with high affinity to HER2, including a HER2 arm identical to Herceptin, a HER2 arm binding a membrane distal epitope termed "153", and a HER2 arm binding a membrane proximal epitope termed "169" (Table 1), all effectively localized to BT474 tumoroids in a dose-dependent manner (Fig. 2A). Conversely, a bsAb binding with lower affinity to the same HER2 epitope as Herceptin (LbD2) showed weak tumoroid localization only at the highest concentrations (0.5 and 1 μg/mL). No localization was observed for a bsAb where the HER2 arm targeted an irrelevant epitope (ctrl). Variations in the CD3 arm (wt, low, ctrl) did not have a significant effect on the localization of bsAbs. We next assessed bsAb accumulation to the outer layers versus penetration into the tumoroids

using higher-resolution confocal images in the center plane of the tumoroids. These images showed that bsAbs with HER2 arms Herceptin, 153, or 169 did not accumulate selectively at the tumoroid border but all penetrated the tumoroids (Fig. 2B,C).

Together, these results showed that HER2 targeting bsAbs readily penetrate the collagen network and subsequently localize to and infiltrate HER2+ tumoroids in a manner depending on their affinity for HER2.

**Screening a series of bsAbs for their efficacy in triggering T-cell recruitment to- and killing of ECM embedded tumoroids**
To address specificity of the bsAb-induced T-cell mediated cytotoxicity analyzed in this setup, we validated the requirement for the specific interaction with the TAA on the tumoroids. Arrays of ECM embedded BT474 tumoroids were generated and exposed to increasing concentrations of

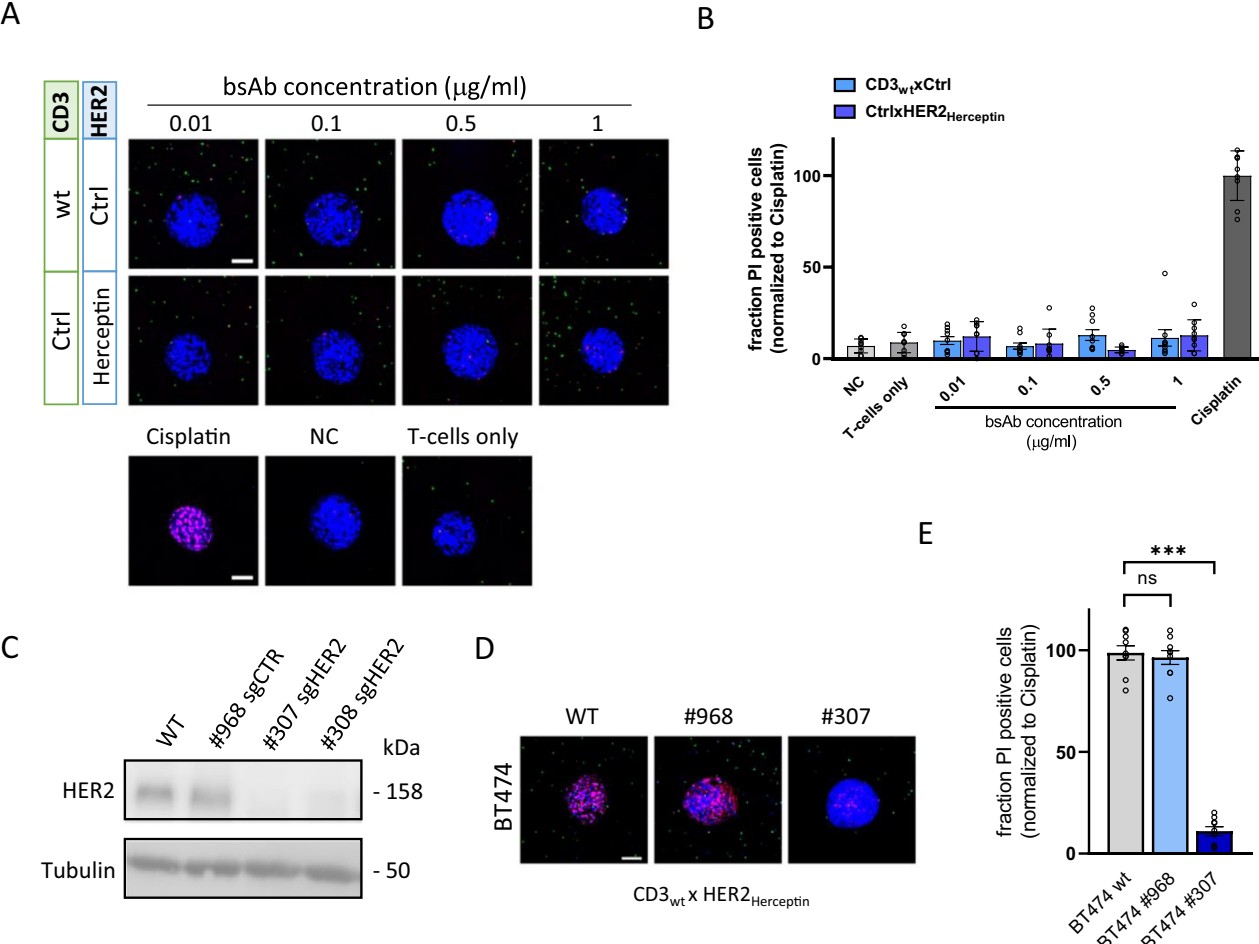

**Fig. 3 | BsAb activity depends on affinity of CD3 and HER2 arms and expression of TAA on tumor cells. A** Maximum projection images showing tumoroid (blue), T-cells (green) and loss of viability detected by PI (red) 6 days after exposure of collagen embedded BT474 tumoroids to a mixture of T-cells and bsAbs with either non-targeting CD3 arm or non-targeting HER2 arm. Bottom panel shows tumoroids treated with 10 μg/mL Cisplatin triggering near complete tumoroid killing versus tumoroids grown in absence of T-cells and bsAbs (negative control; NC) or T-cells only. Bar = 100 μm. **B** Quantification of image data as shown in **A**. Data were normalized to cisplatin condition. Graphs show mean and SEM of 3 independent experiments, each performed in triplicate (3 individual wells each containing 1 collagen embedded BT474 tumoroid). **C** Western blot showing loss of HER2 upon CRISPR-Cas9 mediated knockout in BT474 cells. Tubulin serves as a loading control. **D** Representative maximum projection images at 6 days after exposure of collagen embedded WT, shCTR, or sgHER2 BT474 tumoroids to a mixture of T-cells and 1 μg/mL CD3$_{wt}$xHER2$_{Herceptin}$ bsAbs. Blue, tumor nuclei; green, T-cells; red, PI. bar = 100 μm. **E** Quantification of image data as shown in **D**. Data were normalized to cisplatin condition. Graph shows mean and SEM of 3 independent experiments, each performed in triplicate. *P*-value calculated using two-way ANOVA followed by Bonferroni's multiple comparisons test. ns non-significant; ***$P < 0.001$.

bsAbs in the presence of human healthy donor-derived T-cells. Tumoroids grown in the absence of bsAbs served as negative control, while tumoroids treated with 10 μg/mL cisplatin served as positive control (Fig. 3A, B). Based on pilot experiments determining the optimal timing, T-cell mediated tumoroid killing was assessed at day 6 post T-cell addition by automated confocal imaging of the tumoroid in each well followed by the 3D image analysis pipeline described above. BsAbs with ctrl arms replacing the CD3 or HER2 arm (Table 1) failed to induce T-cell-mediated cytotoxicity (Fig. 3A, B). Furthermore, conditional knockout of *ERBB2* in BT474 cells rendered these cells resistant to T-cell mediated cytotoxicity in the context of a CD3$_{wt}$xHER2$_{Herceptin}$ bsAb binding with high affinity to both CD3 and HER2 (Fig. 3C–E).

We screened a series of CD3xHER2 bsAbs with different CD3 and/or HER2 binding arms varying in affinity and epitope recognition properties (Table 1) for their capacity to kill HER2+ tumoroids. T-cells or bsAbs tested separately did not cause tumoroid killing beyond background levels (Fig. 4B). CD3$_{wt}$xHER2$_{Herceptin}$ bsAbs induced robust T-cell mediated cytotoxicity, with maximal tumoroid killing occurring at 0.1 μg/mL (Fig. 4A, B). When one of the binding arms was replaced with a low affinity

variant (i.e., CD3$_{wt}$xHER2$_{LbD2}$ or CD3$_{Low}$xHER2$_{Herceptin}$ bsAbs), the efficacy decreased. Low affinity at the HER2 arm could be compensated for by increasing bsAb concentration. Low affinity at the CD3 arm could only be partially compensated for in the context of a high affinity HER2 arm at the highest bsAb concentration of 1 μg/mL. BsAbs with a combination of low affinity CD3 and HER2 arms showed no discernible effect on T-cell cytotoxicity across all concentrations tested.

Comparing bsAbs binding with high affinity to CD3 as well as HER2 but interacting with a membrane proximal HER2 epitope CD3$_{wt}$xHER2$_{169}$ or a membrane distal HER2 epitope CD3$_{wt}$xHER2$_{153}$ revealed a major impact of the epitope location (Fig. 4A, B). The CD3$_{wt}$xHER2$_{169}$ bsAb potently induced T-cell mediated cytotoxicity and showed a dose-response resembling that of the CD3$_{wt}$xHER2$_{Herceptin}$ bsAb. By contrast, the CD3$_{wt}$xHER2$_{153}$ bsAb failed to induce T-cell mediated cytotoxicity at any of the concentrations tested. Notably, this difference was observed despite the fact that bsAb recruitment to- and penetration of the tumoroids was identical (Fig. 2). As observed with the CD3$_{Low}$xHER2$_{Herceptin}$ bsAb, a low affinity CD3 arm combined with the 169 HER2 arm was still partially effective at the highest bsAb

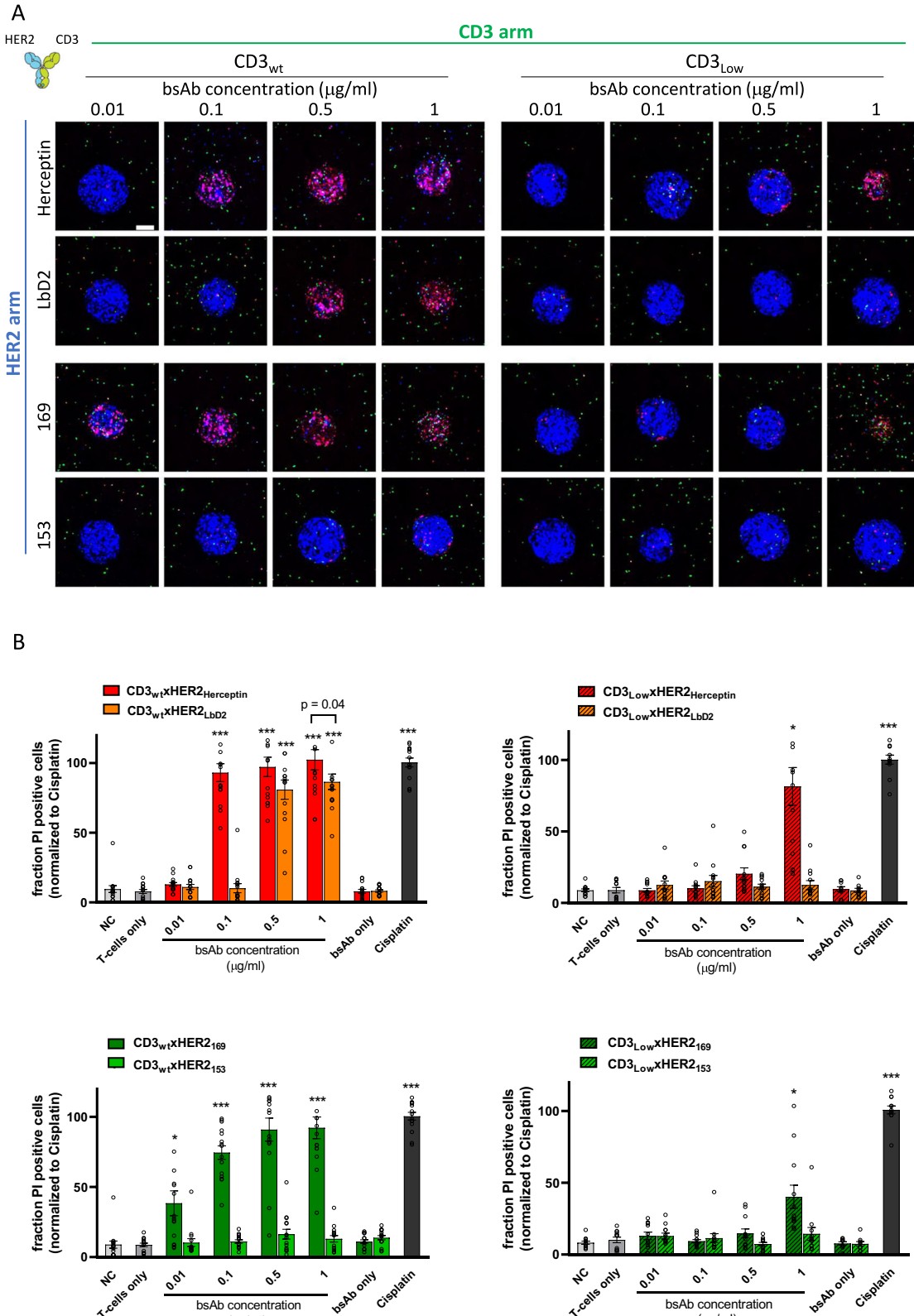

**Fig. 4 | Results of bsAb screening on ECM embedded BT474 tumoroid – T-cell co-cultures.** **A** Maximum projection images showing tumoroid (blue), T-cells (green) and loss of viability detected by PI (red) 6 days after exposure of collagen embedded BT474 tumoroids to a mixture of T-cells and bsAbs for one representative experiment of at least 4 biological replicates. Blue, tumor nuclei; green, T-cells; red, PI. Bar = 100 µm. **B** Quantification of the image data as shown in **A**. Tumoroids grown in absence of T-cells and bsAbs are shown as negative control (NC). Tumoroids treated with 10 µg/mL Cisplatin are shown as positive control triggering near complete tumoroid killing. Results are normalized to cisplatin condition. Graphs show mean and SEM of 5 (CD3$_{wt}$xHER2 variants) or 4 (CD3$_{Low}$xHER2 variants) independent experiments, each performed in triplicate (3 individual wells each containing 1 collagen embedded BT474 tumoroid). *P*-value calculated using two-way ANOVA followed by Bonferroni's multiple comparisons test. *$P < 0.05$; **$P < 0.01$; ***$P < 0.001$ compared to NC.

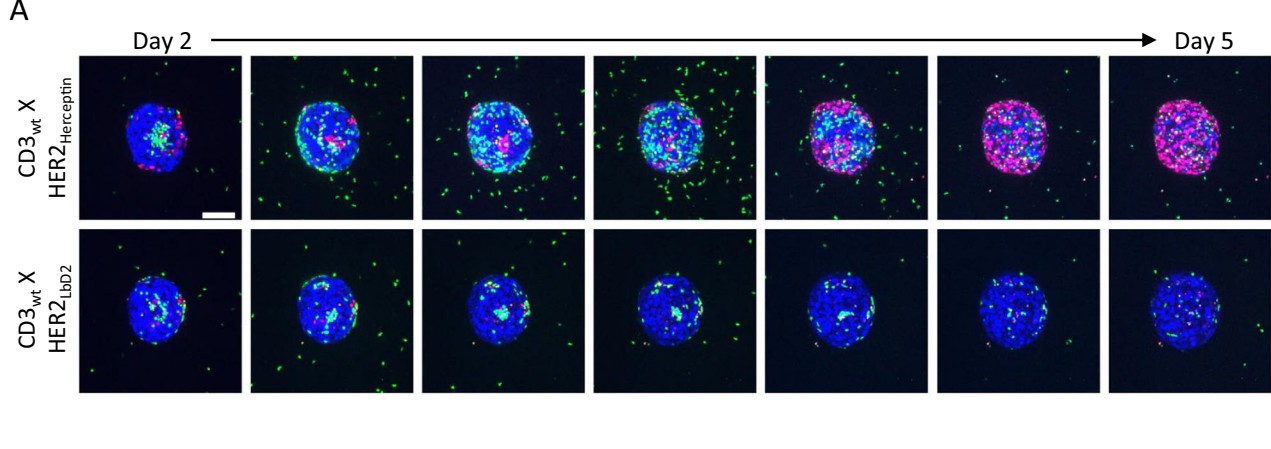

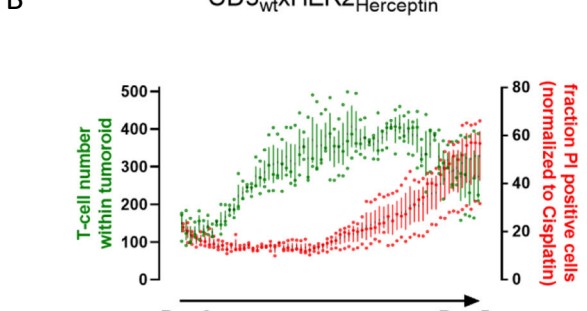

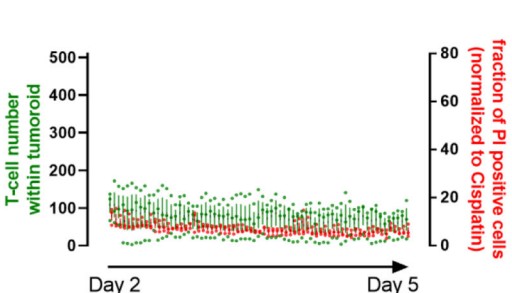

**Fig. 5 | Dynamics of T-cell recruitment and tumoroid killing for effective versus non-effective bsAbs. A** Maximum projection images showing T-cell recruitment to the tumoroid and tumor killing with 12 h intervals from day 2 to day 5 after exposure of collagen embedded BT474 tumoroids to a mixture of T-cells and 0.1 μg/mL of the indicated bsAbs. Blue, tumor nuclei; green, T-cells; red, PI. Bar = 100 μm.

**B** Quantification of time-lapse image data with 1-h intervals as shown in **A**. Green line represents the number of T-cells localized in the tumoroid. Red line represents the percentage of PI positive tumor cells, indicating loss of viability. Mean and SEM of three tumoroids is shown.

concentration (1 μg/mL). The $CD3_{Low}xHER2_{153}$ bsAb was not effective at any of the concentrations tested.

In summary, these results demonstrated the application of the developed ECM embedded tumoroid arrays to screening of bsAbs with variations in their TAA or CD3 binding arms. T-cell mediated cytotoxicity was strictly dependent on expression of the TAA, and the efficacy was modulated by the affinity of the interaction with the CD3 and the TAA. Lower affinity at the TAA binding arm could be compensated for by increasing bsAb concentrations. For low affinity interactions with CD3 such compensation was less effective. The location of the epitope recognized on the TAA appeared to be a critical determinant of the efficacy of T-cell-mediated tumor killing. A bsAb binding a membrane distal HER2 epitope failed to induce tumoroid killing at all concentrations tested, despite the fact that the interaction is high affinity and promotes tumor cell killing in 2D co-culture assays.

**Correlation between T-cell recruitment dynamics and tumoroid killing**

In addition to the assessment of T-cell mediated cytotoxicity, the 3D ECM embedded tumoroid arrays where T-cells are added on top of the collagen gels, allowed analysis of T-cell recruitment towards the tumoroid. We explored the relationship between recruitment of T-cells to the tumoroid and subsequent T-cell mediated tumoroid killing. I.e., we asked whether differences in the ability to trigger tumoroid killing were related to different T-cell recruitment activities or, alternatively, were related to differences in T-cell mediated cytotoxicity of similarly recruited T-cells. First, we evaluated the effect of the HER2 arm affinity on recruitment and subsequent killing. To this end, we compared $CD3_{wt}xHER2_{Herceptin}$ and $CD3_{wt}xHER2_{LbD2}$ bsAbs at 0.1 μg/mL, which triggered effective T-cell mediated tumoroid

killing for Herceptin but no killing for LbD2 bsAbs (Fig. 4). Tumoroids were monitored by automated confocal microscopy from day 2 post T-cell/ bsAb exposure for a period of 72 h capturing images every hour (Fig. 5A; Supplementary Movie 1). Calculation of T-cell recruitment and tumoroid killing using the automated image analysis pipeline described above indicated a similar initial presence at day 2 of ~100 T-cells in tumoroids subjected to $CD3_{wt}xHER2_{Herceptin}$ and $CD3_{wt}xHER2_{LbD2}$ bsAbs (Fig. 5B). For the Herceptin bsAb this initial interaction was followed by a significant increase of T-cell recruitment during the next 2 days followed by a gradual decrease of T-cell localization as tumoroid killing emerged during days 4–6 (Fig. 5A, B). By contrast, tumoroids exposed to the LbD2 bsAb did not show further recruitment of T-cells and no subsequent T-cell mediated cytotoxicity.

Next, we compared $CD3_{wt}xHER2_{169}$ and $CD3_{wt}xHER2_{153}$ bsAbs where affinities for HER2 are high but the distinct locations of the HER2 epitope caused effective (169) or ineffective (153) T-cell mediated cytotoxicity (Fig. 4). In these experiments, plates were slightly tilted during collagen gel polymerization, tumoroids were printed under the highest collagen surface, and T-cells and bsAbs were subsequently added on top of the lower side so the majority of the T-cells would need to actively change direction to reach the tumoroid (Fig. 6A cartoon). Tumoroids were monitored by automated confocal microscopy from day 2 post T-cell/ bsAb exposure for a period of 72 h, capturing images every hour. A few T-cells were observed to have reached the tumoroids in presence of either of the bsAbs at day 2 (Fig. 6A left column; Supplementary Movie 2). However, only $CD3_{wt}xHER2_{169}$ triggered a phase of massive T-cell recruitment at days 3 and 4 whereas $CD3_{wt}xHER2_{153}$ did not lead to any further T-cell recruitment (Fig. 6A). This difference in T-cell recruitment, again correlated

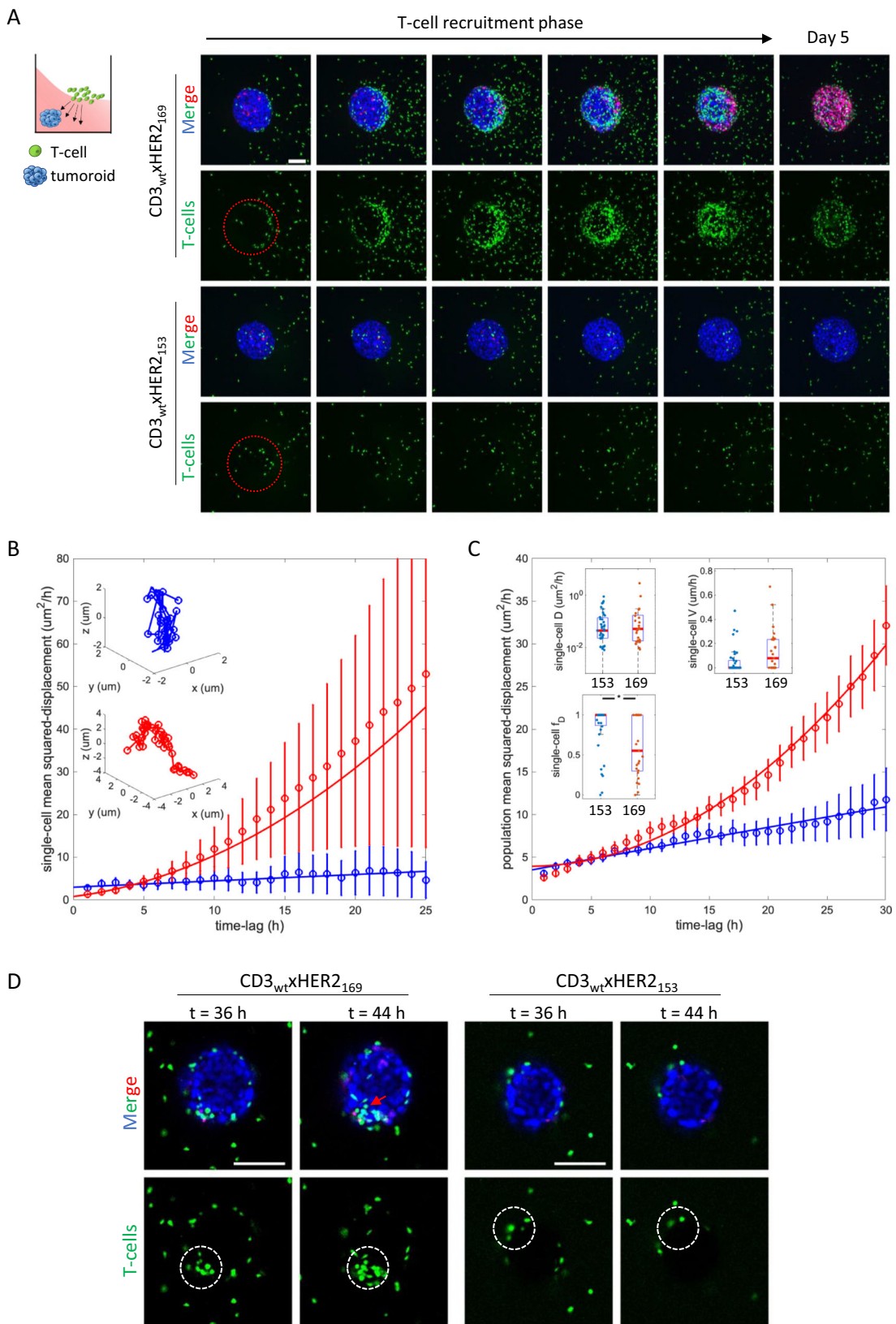

with the difference in subsequent T-cell mediated tumoroid killing (Fig. 6A, PI staining right column).

In order to characterize how application of these bsAbs affected the mode of T-cell mobility, we tracked individual T-cells in the vicinity of the tumor. We assumed a model, in which T-cell mobility is described by an active directional motion, overlaid by a random, diffusive motion. A fit of the mean squared-displacement (MSD) on varying the time-lag between two observations to the respective model (see Methods) yielded the diffusion constant D and the persistent speed V for each of the trajectories observed. For a typical T-cell moving in the presence of

**Fig. 6 | Recruitment of T-cells and tumoroid killing triggered by CD3$_{wt}$xHER2$_{169}$ versus CD3$_{wt}$xHER2$_{153}$ bsAbs. A** Cartoon illustrating experimental setup: BT474 tumoroids were printed below the higher surface of collagen gels with a diagonal upper surface. A mixture of T-cells and bsAbs was added on top of the lower surface. Maximum projection images are shown for bsAbs with a CD3 high affinity arm in combination with a high affinity HER2 arm either recognizing epitope 169 or epitope 153. BsAbs were used at 1 µg/mL. Time lapse starts upon initial contact of T-cells with tumoroids (indicated by red circle; 36 h after adding T-cells and bsAbs). Confocal image stacks were captured every hour. A time span of 40 h is shown with an 8-h interval, followed by an image displaying the final time point (day 5) where T-cell mediated tumoroid killing is near complete. Blue, tumor nuclei; green, T-cells; Red, PI. **B, C** 3D T-cell tracking analysis of time-lapse image data as shown in **A**. **B** MSD analysis of two typical T-cells in the vicinity of the tumoroid. The MSD was determined for time-lag from 1 to 25 h. Red color indicates the presence of CD3$_{wt}$xHER2$_{169}$ and blue color indicates CD3$_{wt}$xHER2$_{153}$. Insets depict the 3D

track of the two T-cells. The MSD for the two cells were characterized by $D_B = 0.x \pm 0.x$ um$^2$/h, $V_B = 0.x \pm 0.x$ um/h, and $D_R = 0.x \pm 0.x$ um$^2$/h, $V_R = 0.x \pm 0.x$ um/h, respectively. **C** MSD analysis of the total population of T-cells in the vicinity of the tumoroid over increasing time-lag from 1 to 30 h. In total 25 and 35 trajectories were analyzed for CD3$_{wt}$xHER2$_{169}$ and CD3$_{wt}$xHER2$_{153}$, respectively. Insets depict median and SD for the parameters diffusion (D), velocity (v), and diffusive fraction ($f_D$) for the population analysis for the indicated bsAbs. Note that $f_D$ is lower for CD3$_{wt}$xHER2$_{169}$ bsAb. *P*-value calculated using multi comparison Dunn's test following non-parametric Kruskal–Wallis test. *$P < 0.05$. **D** Confocal images of a single z-section through the center of a tumoroid exposed to T-cells and either CD3$_{wt}$xHER2$_{169}$ or CD3$_{wt}$xHER2$_{153}$ bsAbs. Initial contact of T-cells with the tumoroid (36 h after adding T-cells and bsAbs) and the same area 8 h later is indicated by the white circle. Red arrow indicates T-cell infiltration occurring only with the CD3$_{wt}$xHER2$_{169}$ bsAb. Blue, tumor nucleus; red, PI; green, T-cell. Bar = 100 µm.

CD3$_{wt}$xHER2$_{169}$, the MSD increased with increasing time-lag whereas for a T-cell moving in the presence of CD3$_{wt}$xHER2$_{153}$ this was not observed (Fig. 6B). The aggregated results for all T-cells observed, showed the same behavior for the MSD and indicated that D was similar in the context of either of these bsAbs while V appeared to increase for T-cells in the presence of CD3$_{wt}$xHER2$_{169}$ even though this did not reach significance (Fig. 6C). Indeed, the diffusive fraction $f_D$ at a time-lag of 10 h ($f_{D10}$) was significantly lower in the presence of CD3$_{wt}$xHER2$_{169}$ indicating that mobility of T-cells in the presence of CD3$_{wt}$xHER2$_{169}$ was dominated by active, directed motion whereas diffusion dominated mobility of T-cells in the presence of CD3$_{wt}$xHER2$_{153}$.

To further analyze T-cell behavior during the recruitment phase at higher time resolution, we used the same experimental setup as in Fig. 6A and performed 8 h time-lapse imaging using a single z-plane through the center of the tumoroid capturing images every 1 min, starting at day 2 when the first T-cells had arrived at the tumoroid. Initial T-cell interactions with the tumoroid border were followed by extensive T-cell dynamics, recruitment of additional T-cells, and invasion of the tumoroid in the context of the CD3$_{wt}$xHER2$_{169}$ bsAb (Fig. 6D; Supplementary Movie 3A). By contrast, the initial T-cell interactions did not cause any further T-cell recruitment or T-cell tumoroid invasion in the presence of the CD3$_{wt}$xHER2$_{153}$ bsAb (Fig. 6D; Supplementary Movie 3B).

Together, these results indicated that successful tumoroid killing is preceded by a wave of T-cell recruitment following initial T-cell tumoroid interactions. Scenarios where bsAbs failed to trigger T-cell mediated tumoroid killing were associated with an inability to trigger such enhanced T-cell recruitment after initial T-cell tumoroid engagement. In addition to affinity for HER2, HER2 epitope location appeared to be a critical determinant of the ability of bsAbs to trigger this wave of T-cell recruitment and subsequent tumoroid killing by T-cells.

## Initial bsAb mediated T-cell-tumor cell interactions trigger chemotactic attraction of additional T-cells

As an explanation for the wave of T-cell recruitment following initial T-cell-tumoroid interactions, which appeared critical for subsequent bsAb-induced T-cell mediated tumoroid killing, we considered the emergence of chemotactic signaling. To address this possibility, tumor cells were seeded in the lower chamber of transwell plates, with or without the addition of unlabeled T-cells, and with or without CD3$_{wt}$xHER2$_{169}$ bsAbs. Cell tracker CMFDA-labeled T-cells were added to the upper chamber, and the number of fluorescent T-cells that migrated to the lower chamber was measured after 48 h (Fig. 7A). In the presence of the positive control chemoattractant CXCL12, migration of labeled T-cells into the lower chamber was observed (Fig. 7B). BT474 cells seeded in the lower chamber did not promote T-cell migration by themselves and minimal T-cell migration was detected when BT474 and T-cells were co-cultured in the lower chamber in the absence of bsAbs. However, co-culture of BT474 and T-cells in the presence of the CD3$_{wt}$xHER2$_{169}$ bsAb promoted extensive T-cell recruitment from the

upper chamber (Fig. 7B bottom right image). Brightfield images showed that this condition caused T-cell clustering and tumor cell killing (Fig. 7B top right image).

Cells were collected from the lower chamber, stained with CD3 antibody, and analyzed by flow cytometry (Fig. 7C). Cells in quadrant 1 (Q1) represented the unlabeled T-cells initially added to the lower chamber. While T-cell numbers in Q1 decreased slightly in absence of bsAbs, the T-cell number increased >1.5-fold in the 48 h time frame in the presence of tumor cells and the bsAb, indicating that T-cells proliferated (Supplementary Fig 2A). Co-culture of BT474 and T-cells in the presence of the CD3$_{wt}$xHER2$_{169}$ bsAb, but not any of the other conditions, led to an increase in forward scatter (FSC) and side scatter (SSC) in Q1 pointing to T-cell activation (see refs. [37,38]), which was confirmed by CD69 staining (Supplementary Fig 2B, C). A weak sign of activation was also observed in tumor cell-T-cell co-cultures in absence of bsAbs, in line with an earlier observation[30]. The T-cells that were recruited to the lower chamber (Q2) by the tumor-, T-cell, bsAb mix showed an activated phenotype based on the shift in FSC/SSC (Supplementary Fig 2D). Notably, this was also observed for the few T cells that were recruited by the tumor-T-cell coculture in absence of bsAbs. These T-cells also showed a decrease in CMFDA fluorescence indicating proliferation, which was not observed for the CXCL12 condition (Supplementary Fig 2E). Importantly, when BT474 and T-cells were co-cultured in the bottom compartment in the presence of the CD3$_{wt}$xHER2$_{169}$ bsAb, the number of recruited T-cells in Q2 was significantly increased as compared to all the conditions without bsAbs, reaching a similar level of T-cell recruitment as triggered by CXCL12 (Fig. 7D, E).

To further establish the involvement of chemotactic signals, conditioned media were collected from the lower chamber for the different conditions tested. For activation of T-cells, CD3 bsAbs require crosslinking through interaction with their other arm with the TAA on tumor cells. Hence, their presence in the conditioned media per se, would not trigger T cell recruitment. Conditioned medium obtained from a 24 h co-culture of BT474 and T-cells in the presence of the CD3$_{wt}$xHER2$_{169}$ bsAb promoted significant attraction of T-cells from the upper chamber pointing to the presence of chemotactic factors (Fig. 7F; Supplementary Fig 3). Such activity was not observed in absence of the bsAb and was lost when conditioned medium was collected at 48 h, which may be due to excessive tumor cell death. Finally, while the CD3$_{wt}$xHER2$_{153}$ bsAb did not stimulate T-cell mediated killing of ECM embedded HER2+ tumoroids (Fig. 4), in 2D co-culture it effectively triggered killing of BT474 cells (Fig. 7G). Moreover, in 2D co-culture the CD3$_{wt}$xHER2$_{153}$ bsAb also stimulated recruitment of T-cells from the upper chamber indicating that chemotactic factors were produced (Fig. 7H) whereas in the context of 3D ECM tumoroids this bsAb failed to trigger T-cell recruitment to- and killing of the tumoroid (Fig. 6; Supplementary Movie 2; Supplementary Movie 3).

In summary, these findings confirmed that bsAb mediated T-cell-tumor cell interactions trigger chemotactic signaling. This may underlie the second phase of T-cell recruitment observed in 3D ECM embedded tumoroid co-cultures, which we identify as a crucial aspect of the efficacy of

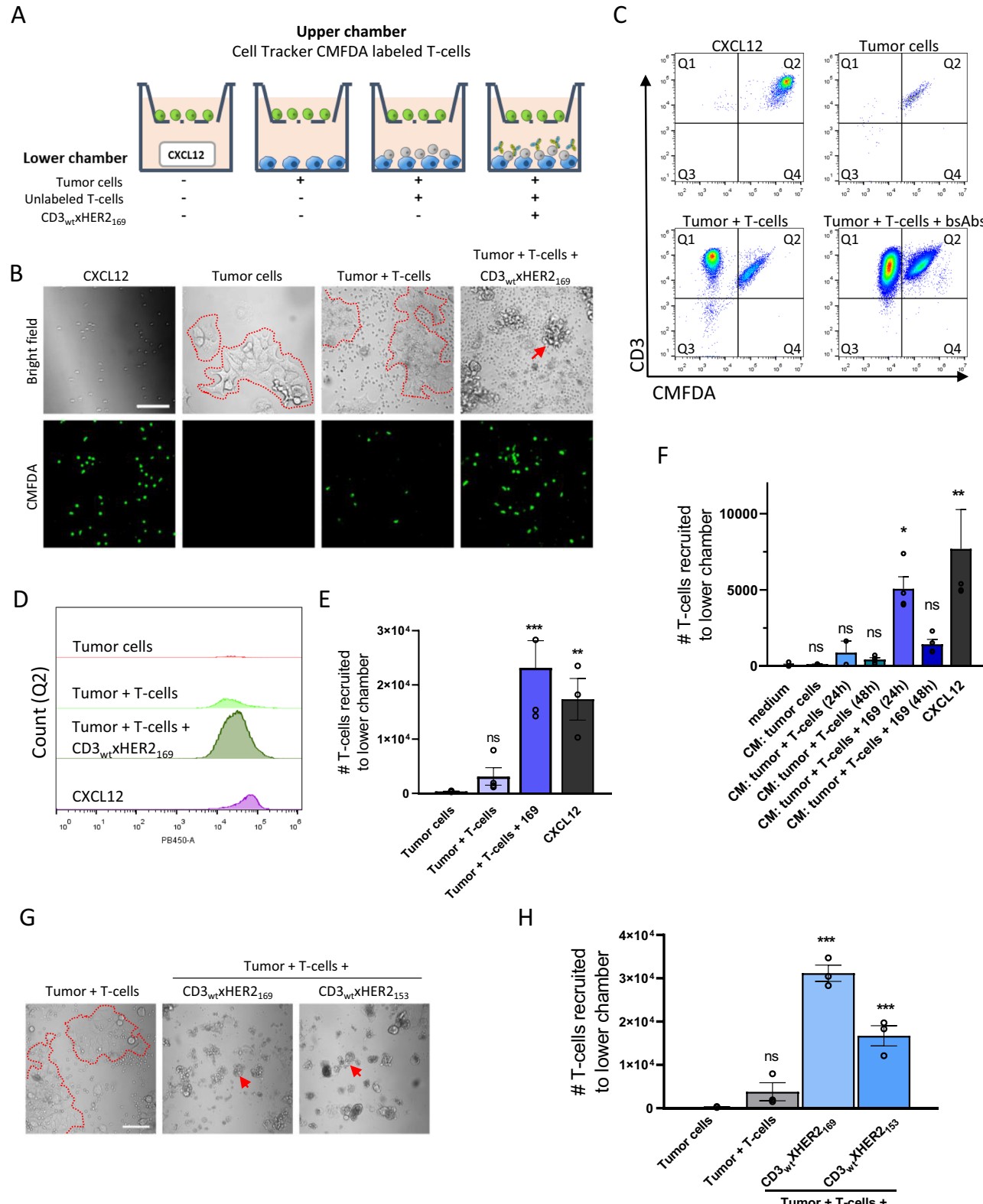

bsAbs. Although EC50s for HER2$_{169}$ and HER2$_{153}$ containing bsAbs differ in 2D co-culture assays, the fact that the CD3$_{wt}$xHER2$_{153}$ bsAb supports T-cell mediated killing of a 2D tumor cell monolayer culture and in that setup also triggers chemotaxis, indicates that bsAb evaluation in 2D co-culture assays fails to predict efficacy in the context of a 3D tumor structure within an ECM.

## Discussion

BsAbs have entered the clinic for treatment of liquid malignancies but their use for the treatment of solid tumors remains challenging[15,16,20]. For carcinomas, including breast cancer, combinations with other strategies such as chemotherapy or checkpoint inhibitors may be a way forward, as well as exploration of new targets and further optimization of existing bsAbs[22,39].

**Fig. 7 | BsAb mediated T-cell-tumor cell engagement triggers chemotactic signals recruiting additional T-cells. A** Schematic illustration of the transwell assay. BT474 cells were seeded in the lower chamber of a transwell plate, with or without the addition of unlabeled T-cells and with or without CD3$_{wt}$xHER2$_{169}$ bsAbs. Cell Tracker CMFDA-labeled T-cells were added to the upper chamber. The number of green-fluorescent T-cells migrating to the lower chamber was analyzed after 48 h using confocal microscopy and flow cytometry. A condition using 100 ng/mL CXCL12 was used as positive control. **B** Bright field images (showing a mixture of unlabeled T-cells, infiltrating CMFDA-labeled T-cells, as well as tumor cells) and green fluorescence channel images (showing labeled T-cells) taken in the lower chamber at 48 h for the indicated conditions. Red dotted line outlines islands of tumor cells. Red arrow indicates a T-cell cluster. Bar = 100 μm. **C** Flow cytometry analysis of cell populations from the lower chamber at 48 h. Horizontal axis shows CMFDA signal (negative for unlabeled T-cells co-cultured with tumor cells in bottom chamber; positive for CMFDA-labeled T-cells recruited from the upper chamber); vertical axis shows CD3 levels. Flow cytometry analysis (**D**) and bar graph showing mean and SEM from 3 to 4 biological replicates (**E**), depicting cell counts in

Q2 (CMFDA:CD3 double positive T-cells migrated from the upper to the lower chamber) for the indicated conditions. *P*-value calculated using one-way ANOVA followed by Bonferroni's multiple comparisons test. ns, non-significant; **$P < 0.01$; ***$P < 0.001$. **F** T-cell recruitment triggered by conditioned media (CM) from the indicated conditions. Graph shows cell counts of CMFDA-labeled infiltrated T-cells collected from the lower chamber. Mean and SEM from 2 to 4 biological replicates. *P*-value calculated using one-way ANOVA followed by Bonferroni's multiple comparisons test. ns non-significant; *$P < 0.05$; **$P < 0.01$. **G** Bright field images (showing a mixture of unlabeled T-cells, infiltrating CMFDA-labeled T-cells, as well as tumor cells) taken in the lower chamber at 48 h for the indicated conditions. Red dotted line outlines islands of tumor cells. Red arrows indicate T-cell clusters. Bar = 100 μm. **H** Bar graph showing mean and SEM from 3 biological replicates, depicting cell counts in Q2 (CMFDA:CD3 double positive T-cells migrated from the upper to the lower chamber) for the indicated conditions. *P*-value calculated using one-way ANOVA followed by Bonferroni's multiple comparisons test. ns non-significant; **$P < 0.01$.

Screening of new bsAbs, before entering preclinical in vivo testing, typically relies on 2D co-cultures, which poorly recapitulate biomechanical and biochemical properties of solid tumors[23–25]. Mixed 3D co-cultures of tumor spheroids or organoids with T-cells have advanced the field but do not allow monitoring of recruitment of T-cells from the environment towards the tumoroid[31–33]. The platform we describe here, permits high-throughput, automated confocal microscopy-based screening of T-cell activity against arrays of 3D solid tumoroids. The tumoroids are embedded in a fibrillar collagen-based ECM resembling the collagen-rich ECM of the stroma into which solid tumors, including breast cancers grow and invade[40]. The assay is accompanied by an automated 3D image analysis pipeline providing quantitative T-cell recruitment and tumoroid killing data for (antibody) drug screening.

We have used this setup to monitor over a timeframe of ~1 week how i) freshly isolated non-activated human T-cells travel through a fibrillar collagen-based ECM surrounding the tumoroid, ii) how these T-cells make initial bsAb-dependent contact with the tumor, iii) how this triggers a surge of further T-cell recruitment from the surrounding ECM, and iv) how this ultimately drives a full-blown immune attack causing tumor destruction. Monitoring these different phases in an in vitro drug screening format has not been established before. In earlier studies T-cells were pre-mixed[41] or added to "naked spheroids" formed by hanging-drop methodology[33,42], to study their intratumor migration and tumor killing capacity. An interesting, albeit lower throughput method has made use of bioprinting with alginate-, gelatin-, and Matrigel-based bioinks to separate a tumor organoid from a surrounding layer of T-cells[43] but this does not recapitulate interactions in a fibrillar stroma-like ECM.

We implemented this platform in the testing of a series of bsAbs combining a CD3 and a HER2 binding arm to redirect T-cells to HER2+ breast cancer cells. Our observation that higher affinity for HER2 enhances tumoroid killing is in line with a report that high affinity HER2 binding is critical for efficacy of CD3xHER2 bsAbs in a preclinical orthotopic breast cancer model[44]. However, in the same study, high affinity HER2 binding also leads to increased toxicity through cytokine release and detrimental effects on HER2 expressing tissues, indicating that affinity at the HER2 arm must be carefully finetuned. An alternative strategy has been reported where two low affinity HER2 arms are combined to optimize selectivity against HER2 amplified tumor cells[45]. We find that lower affinity on the HER2 binding arm is readily compensated for by increasing bsAb concentrations but the impact of CD3 affinity appears to be more pronounced, as increasing bsAb concentrations only partially compensates for a low affinity CD3 arm. There is some discrepancy between reports evaluating the impact of CD3 affinity on bsAb efficacy in 2D co-cultures. Lowering CD3 affinity either reduced[46], enhanced[47,48], or had no discernable effect on bsAb mediated T-cell cytotoxicity[44]. Importantly, preclinical in vivo studies using CD3xHER2 bsAbs showed that too high affinity for CD3 shifts the distribution from

HER2+ tumors to secondary lymphatic tissues[22] and leads to increased toxicity through cytokine release[44].

Besides affinity, we find that the location of the TAA epitope is another critical determinant of bsAb efficacy. These findings extend prior reports that bsAbs targeting membrane proximal epitopes in the FcRH5 surface marker on myeloma cells or in the melanoma-associated chondroitin sulfate proteoglycan (MCSP) on melanoma cells, outperform those targeting membrane distal epitopes[49,50]. Our results indicate that this plays a crucial role particularly in the context of an ECM embedded 3D tumor micro-environment: i.e., even though a high affinity bsAb recognizing a membrane distal HER2 epitope i) supports T-cell mediated killing of tumor cells in 2D co-culture and ii) readily concentrates in an ECM embedded tumoroid; it fails to promote active recruitment of T-cells and subsequent T-cell mediated tumoroid destruction, and this cannot be compensated for by increasing bsAb concentrations. Altogether, these studies highlight the importance of careful tuning of the affinities and epitope locations to develop bsAbs with favorable efficacy versus toxicity profiles. It is to be expected that any optimization of bsAb efficacy is related to patient stratification, with for instance distinct requirements for CD3xHER2 bsAbs in the context of HER2 amplified versus HER2 low cancers. Our data also indicate potential chemotactic recruitment of T-cells to immunologically cold tumors, provided that some initial engagement of a population of already present T-cells with TAA positive tumor cells is triggered by bsAbs. Thus, cold tumors harboring a very low percentage of T-cells or immune-excluded cold tumors with T-cells present at the edge of the tumor may benefit from bsAb therapy by a few initial engagements triggering influx of additional T-cells, whereas this is unlikely for immune-desert cold tumors lacking T cells in the tumor and the tumor microenvironment.

We show that initial bsAb mediated contact of a few T-cells with a 3D cluster of tumor cells embedded within a fibrillar ECM, resembling a tumor microenvironment in which solid tumors, including breast cancers grow and invade, only leads to a full-blown immune attack if a subsequent wave of T-cell recruitment is triggered. Such T-cell redirection toward tumor cells after initial T-cell interactions has been proposed for a CD3xEGFR bsAb in the context of colorectal cancer[51]. Similar behavior has also been recently described by computational modeling for T-cells targeting an MHC/oval-bumin peptide complex expressed in melanoma cells[52]. Our work indicates that such a positive feedback mechanism, triggered by initial T-cell tumor contact is critical for successful bsAb mediated T-cell anti-tumor activity. CD3xHER2 bsAbs that fail to induce T-cell mediated tumoroid killing due to insufficient affinities at either arm or due to binding to a membrane distal HER2 epitope, also fail to trigger recruitment of T-cells from the surrounding ECM after initial T-cell tumoroid interaction. We use a transwell setup to show that T-cell tumor cell interactions indeed trigger paracrine signaling to attract additional T-cells and we find that such chemotactic factors disappear at later stages. Notably, we find that a bsAb that is

ineffective in 3D does in fact mediate T-cell cytotoxicity in 2D co-cultures and in that scenario also triggers paracrine signaling to attract T-cells. This indicates that bsAb testing in 2D co-cultures fails to predict efficacy in the context of a 3D ECM embedded tumor.

In summary, we have established an immuno-oncology screening platform delivering quantitative 3D imaging-based data that can be implemented for assessment of the efficacy of a range of CD3xTAA bsAbs. We use this setup to monitor the interaction of human healthy donor-derived T-cells with ECM embedded breast cancer tumoroids over a ~1 week timeframe in the context of a panel of CD3xHER2 bsAbs. This i) allows identification of bsAbs with optimal epitope locations and affinities in their CD3 and TAA binding arms leading to a full-blown immune attack causing tumoroid destruction and ii) demonstrates that effective bsAbs trigger a surge of chemotactic T-cell recruitment following initial T-cell tumoroid interaction.

## Methods
### Cells and reagents
Human breast cancer cell lines BT474 and MDA-MB-231 were obtained from American Type Culture Collection (ATCC), authenticated using short tandem repeat (STR) profiling, and tested negative for mycoplasma prior to use. Cells were grown in RPMI 1640 (52400-025, Gibco, Fisher Scientific, Landsmeer, The Netherlands) supplemented with 10% fetal bovine serum (FBS), 25 U/mL penicillin, and 25 µg/mL streptomycin in a humidified incubator with 5% $CO_2$ at 37 °C. T-cells were purified from buffy coats of healthy donors (Sanquin, Amsterdam) by Ficoll Paque Plus (GE17-1440-02, Merck). CD3-positive T-cells were subsequently isolated using the Pan T-cell Isolation Kit (130-096-535, Miltenyi Biotec) following the manufacturer's instructions. In brief, peripheral blood mononuclear cells were incubated with a cocktail of biotin-conjugated non-T-cell monoclonal antibodies for 10 min at 4 °C. Subsequently, anti-biotin monoclonal antibodies conjugated to magnetic MicroBeads were added to bind non-T-cells for 15 min at 4 °C. The cell suspensions were applied to MACS columns and a MACS Separator, allowing the magnetically labeled non-T-cells to be separated from the fraction of CD3-positive T-cells. The unlabeled, enriched T-cells were collected and confirmed by CD3 flow cytometry before utilization.

Tumor cells were stained overnight at 37 °C with 1 µg/mL Hoechst33342 (#610959; Thermo Fisher) before being used for generation of ECM embedded tumoroids. 1 µM CellTracker CMFDA (C2925; Thermo Fisher) in complete culture medium was used to label T-cells for 20 min at room temperature (RT) before being added to ECM embedded tumoroids. Culture media added on top of ECM embedded tumoroids contained 0.4 µM PI to label dead cells. Cisplatin was obtained from the Pharmacy unit of University Hospital Leiden, Leiden NL and used as a stock concentration of 5 mg/mL to prepare concentration ranges in culture medium. CXCL12 was obtained from R&D systems (350-NS).

### Conditional knockout procedure
Conditional knockout cell lines were generated as described previously[53]. In short, BT474 cells were transduced with Lentiviral Edit-R Cas9 plasmid (Dharmacon) and selected by 2 µg/mL blasticidin. Subsequently, inducible BT474-Cas9 cells were transduced with a non-targeting sgRNA (#968) and two sgRNAs targeting the *ERBB2* gene (Sanger Arrayed Whole Genome Lentiviral CRISPR Library (Sigma-Aldrich); sgHER2 #307, *CCCCAGG-GAGTATGTGAATGCCA*; sgHER2 #308, *CAACTACCTTTCTACG-GACGTGG*) and bulk selected by 4 µg/mL puromycin. HER2 knockout cells were generated by exposure of cells to doxycycline for 7 days followed by negative fluorescence-activated cell sorting (FACS) using a HER2 antibody (1:200; 2165T, Cell Signaling; RRID: AB_10692490) and AlexaFluor-647 conjugated anti-rabbit secondary antibody (1:500; 111-605-003, Jackson; RRID: AB_2338072).

### Western blotting
Cells were lysed with RIPA buffer containing 1% protease/phosphatase inhibitor cocktail (PIC; P8340, Sigma-Aldrich). Samples were separated by SDS-polyacrylamide gel electrophoresis and transferred to polyvinylidene difluoride (PVDF) membranes (Millipore) followed by blocking with 5% BSA in Tris-buffered saline with 0.05% Tween-20. Membranes were incubated with primary antibodies against HER2 (1:500; MA5-14057, Thermo Scientific; RRID: AB_10977723) or tubulin (1:1000; T9026, Sigma-Aldrich; RRID: AB_477593) overnight at 4 °C followed by Horseradish peroxidase (HRP)-conjugated secondary antibodies (Jackson Immunoresearch, anti-rabbit 111-035-003; RRID: AB_2313567; anti-mouse 115-035-003; RRID: AB_10015289) for 1 h at RT, and imaged with enhanced chemiluminescence substrate mixture (ECL plus, Amersham, GE Healthcare, Chicago IL, USA). Blots were imaged using an Amersham Imager (GE, Healthcare Life Science, Chicago, IL, USA).

### Bispecific antibodies
The variable domains of CD3 and HER2 antibodies were cloned in a human IgG1κ backbone containing the L234F/L235E/D265A (FEA; Eu numbering[54]) Fc-silencing mutations and the K409R (HER2 antibody) or F405L (CD3 antibody) mutations. Fc inert bsAbs were generated using controlled Fab-arm exchange, termed DuoBody® technology as previously described[34–36]. Binding to CD3 on healthy donor T-cells and to HER2 on breast cancer cells was confirmed[34]. Properties of bsAbs used in this study are listed in Table 1. To assess their localization in ECM embedded tumoroid cultures, bsAbs were labeled with Alexa Fluor 647.

### ECM embedded tumoroid arrays and exposure to T-cells and bsAbs
Collagen type I solution was isolated from rat tails by acid extraction as previously described[25,55]. Collagen (stock concentration: 5 mg/mL) was diluted to 1.5 mg/mL in DMEM containing 0.1 M HEPES (H0887, Sigma-Aldrich) and 44 mM $NaHCO_3$ (stock 440 mM; 71630, Fluka). 70 ul of collagen solution per well was loaded into a 96-well plate (655090, Greiner) and polymerized for 1 h at 37 °C. Subconfluent monolayers of tumor cells were trypsinized, filtered (04-0042-2317, Sysmex), and re-suspended in 100 µl complete culture medium containing 2% polyvinylpyrrolidone (PVP; P5288, Sigma-Aldrich). Droplets of the PVP/cell suspension containing ~5000 cells were printed into the collagen gels at defined x-y-z positions 150 µm above the bottom of the wells by image guided injection using a micro-injection robot (Life Science Methods, Leiden, NL) as previously described[25,56]. For T-cell co-cultures, ~50,000 CMFDA-labeled T-cells were added on top of the collagen gels in presence or absence of bsAbs.

### Automated imaging of ECM embedded tumoroid arrays and 3D image analysis pipeline
Tumoroids were imaged using a Nikon TE2000 confocal microscope equipped with a prior automated stage controlled by NIS Element Software at 20× objective or 40× long distance water immersion objective in a temperature and $CO_2$ controlled incubator. Either confocal Z stacks were generated through the entire z-axis of the tumoroid taking an image every 10 µm, or a single z-section was captured through the center of the tumoroid. For cisplatin treatment, the tumoroids were captured at 24 hours and 48 h after treatment. For bsAb screening, images were captured at day 6 after T-cell/ bsAb addition. For time-lapse imaging, tumoroids were monitored for 72 h with a 1-h time interval taking confocal images across the entire tumoroid z-axis, or for 8 h with a 1-min time interval at one confocal z-section through the center of the tumoroid.

Automated image analysis was performed using ImageJ 1.53c and CellProfiler version 2.2.0. The images from the Hoechst channel were pre-processed by ImageJ to create masked images and identify the tumoroid boundary using watershed masked clustering (WMC) segmentation. Subsequently, the masked Hoechst images, along with the other channels were imported into CellProfiler for further processing of each individual z-plane as follows:

1. Identify Primary Objects: identify and assign the objects from the blue channel (Hoechst; tumor cells), the red channel (PI; dead cells), the green channel (CellTracker CMFDA; T-cells);

2. Mask Objects: mask T-cell images and remove all T-cells outside the tumoroid boundary;
3. Relate Objects: assign PI positive cells overlapping with tumor cells (parent object) as dead tumor cells (child object);
4. Filter Objects: remove all PI signals that do not overlap with tumor cells;
5. Calculate Math: calculate the fraction of PI positive cells in each plane for a z-stack;
6. Export the data and combine the information in each z-plane for each tumoroid to quantify the total number of T-cells per tumoroid and the fraction of PI positive tumor cells per tumoroid.

Images focused on T-cells in the vicinity of tumoroids were analyzed using a MatLab script (MatLab R2018a; MathWorks, Natick, MA, USA). For each timepoint image stacks where first thresholded in the respective CellTracker CMFDA channel. The 3D center-of-mass positions of all thresholded objects that had the predicted volume of a cell ($280\ um^3 < $ volume $ < 10,000\ um^3$) where determined. From the 3D position data, cell trajectories in time were constructed using an assignment algorithm described earlier[57]. The mobility of each T-cell that was observed for at least n hours was further analyzed in terms of the change in the mean-squared displacement (MSD), with a lag time ($t_{lag}$) between 2 time frames. We considered two types of movement: one involving diffusion, which is described by a diffusion constant (D), and a second describing active, directed motion characterized by a velocity (v). As previously described[57,58] in this situation the MSD changes with time-lag as:

$$msd\left(t_{lag}\right) = 4Dt_{lag} + v^2 t_{lag}^2 \qquad (1)$$

In order to characterize the overall motility, we further defined the diffusive fraction ($f_D$) as the ratio of the diffusive part of the MSD (4D $t_{lag}$) to the total MSD, at a fixed lag time time, $t_D = 10$ h. Hence the diffusive fraction is given by:

$$f_D = \frac{1}{1 + \frac{v^2}{4D} \cdot t_D} \qquad (2)$$

**Transswell T-cell migration assay**

25,000 BT474 cells were seeded per well in the lower chamber of a transwell plate (662630; Greiner Bio-one), in the presence or absence of unlabeled T-cells (125,000 cells/well; effector:target ratio = 5:1) and with or without 1 µg/mL bsAb. As a positive control, 100 ng/mL CXCL12 was added to the lower chamber and in some experiments conditioned media were collected at 24 h and 48 h after tumor – T-cell co-culture and added to the lower chamber after filtration to remove cells in the medium (431220, Corning). Subsequently, 100,000 CellTracker CMFDA-labeled T-cells were added to the upper chamber. After 24 or 48 h, the upper chambers were discarded, lower chambers were imaged capturing bright field and green-fluorescent channels using a Fluorescent Cell Imager (ZOE™; Bio-Rad). Subsequently, the cells from the lower chambers were collected for further characterization of the cell populations by flow cytometry.

**Flow cytometry**

Cells from the lower chamber of a transwell plate were collected and washed twice with FACS buffer (PBS supplemented with 2%FBS, 1 mM EDTA). Subsequently, the cells were incubated with Brilliant Violet 421™ anti-human CD3 antibody (317343; BioLegend, San Diego CA, USA; RRID: AB_2565848) or PE-Cyanine7 anti-human CD69 antibody (310911; Bio-Legend, San Diego CA, USA; RRID: AB_314846) for 20 min at 4 °C. Cell suspensions were again washed twice with FACS buffer and analyzed by flow cytometry (CytoFLEX, Beckman Coulter). Unstained T-cells were used as negative control. To quantify the number of CMFDA-labeled T-cells

migrating to the lower chamber, samples were run until the end to collect all cells. FACS data were analyzed using FlowJo (version 10).

**Statistics and reproducibility**

Statistical analyses were performed in GraphPad Prism 8 using one-way analysis of variance (ANOVA) or two-way ANOVA with Bonferroni's post-hoc test, unless otherwise specified. Data were presented as mean and standard error of the mean (SEM). Statistical significance was considered when $p < 0.05$. Experiments were conducted using three biological replicates, each performed in triplicate, unless otherwise specified. Details are provided in the figure legends.

**Reporting summary**

Further information on research design is available in the Nature Portfolio Reporting Summary linked to this article.

**Data availability**

All data generated or analyzed during this study are included in this published article and its supplementary information files. The source data behind the graphs in the paper can be found in Supplementary DATA.

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

## Acknowledgements

E.H.J.D. and T.S. were supported by a grant from the Dutch Research Council (NWO; Science-XL grant 2019.022). The authors thank Bram Slütter (LACDR, Leiden University) for providing the CD69 antibody.

## Author contributions
E.H.J.D. conceived and supervised the project. C.-Y.L., and E.H.J.D. conceptualized and designed experiments. C.-Y.L. and J.E.K. performed experiments. C.-Y.L. and T.S. analyzed data. P.E., A.I.-F. and R.R. provided bsAbs, advice, and support. C.-Y.L. and E.H.J.D. wrote the manuscript. All authors read and reviewed the manuscript.

## Competing interests
This research was partly funded by Genmab B.V. P.E., A.I.-F. and R.R. are employees of Genmab B.V. All other authors declare no competing interests.
