## [Transparent Peer Review file · Communications Biology]

CD3-engaging bispecific antibodies trigger a paracrine regulated wave of T-cell recruitment for effective tumor killing

Corresponding Author: Dr Erik Danen

Version 0:

Reviewer comments:

Reviewer #1

(Remarks to the Author)

The authors address a novel important mechanism in tumor eradication by T-cell recruitment, using arrays of 3D tumoroids seeded in extracellular matrix. Here they screen for activity a series of bispecific antibodies of effector-silent Duobody format, which are directed against CD3 and Her2, and of variable affinities, and also directed against different epitopes of Her2. They discover that for certain "effective" bsAbs, the human healthy-donor derived T cells can be efficiently recruited after the initial tumor - T cell contact, which results in potent tumor killing. Their evidence is based on quantitative 3D-image analyses of confocal microscopy derived data. Interestingly, they observe not only efficient activation of T cells, but also proliferation after exposure to co-culture of target and T cells, which was not the case when the recruitment was elicited with CXCL12 cytokine.

These findings are highly important for future design of discovery of bispecific T-cell engaging antibodies, and the methodology presented here is very well supported by use of several controls, including low-expressing Her2 cell lines and knock-out control cell lines.

Please find below a list of remarks which I hope you will find helpful.

1. Page 1: „T cell recruitment” – should be T-cell recruitment (please correct throughout the text). T cell is not hyphenated, but expressions such as T-cell recruitment, movement, etc. are hyphenated.
2. Page 1: “despite equal high affinities” – equally high?
3. Page 3: “immune checkpoint inhibitors can improve the response to CD3xHer2 bsAbs” – this statement is too general. Mentioning the opportunities given by involving immune checkpoints into therapeutic approaches with bispecifics might be beneficial.
4. Page 3: *in vitro/in vivo* should be in italics
5. Supplementary Figure 1: “1 collagen embedded BT474 tumoroid” – or MDA-MB231 tumoroid
6. Page 4: “tumor nuclei were detected (Hoechst) and subsequently scored as PI positive or negative” – I suppose the cells were scored as PI positive or negative
7. Page 4: In the figure 2C, the two shades of blue showing DII and DIII of Her2 are difficult to differentiate.
8. Page 4: Fc-inert, not Fc inert. At this point it should be explained which changes were made to obliterate the Fc function.
9. Page 5: 1 ug/mL – micrograms should be correctly denoted, please correct throughout the manuscript
10. Page 5: CD3low: could the CD3 affinity for unmodified and low affinity variant be specified?
11. Page 11: “optimalization of bsAb efficacy is related to patient stratification” – in the further sentence, only Her2 expression on tumor is described, but it would be interesting to comment on the novel findings in view of immunologically cold tumors
12. From page 12: all antibodies used should be listed with RRDs
13. Page 15: Brilliant violet
14. Figure 3C. Western blots for Her2 and the control do not correspond to the same gels, or the discrepancies might be artefactual – please improve on the presentation of the results.
15. Byline to figure 6, page 29: MSD analysis of the total population of T cells in the vicinity of the tumoroid tumoroid over increasing time-lag from 1-30h – tumoroid only once
16. Page 32, legend to Table 1: I could not quite retrieve which of the variants from the reference 58 was used in this study.

Reviewer #2

(Remarks to the Author)

The authors developed a platform, which allows one to generate 3D tumoroids embedded in collagen matrix to which T cells can be added for evaluation of T cell recruitment, activation and tumor cell killing. The authors show that different bispecific anti-Her2-CD3 antibodies display different abilities to kill tumor cells depending on their affinity to Her2 and CD3 and also depending on the location of the targeted Her2 epitopes. They could confirm previous findings that membrane-proximal epitopes elicit higher T cell activation, likely due to the formation of a tighter cytolytic synapse. The authors could trace the migration of single T cells to tumor cells and also provide evidence that chemotactic factor(s) play a role in T cell recruitment and accumulation at the tumoroid. Altogether, these are extremely interesting results and the presented platform allows for high throughput screening of bispecific antibodies for efficient tumor cell killing in a 3D tumor model, which is of great value for therapeutic antibody development. The paper is very well written and the experiments and their statistical analysis are sound and conclusive. The manuscript definitely merits publication.

Minor points:

- 1) Figure 3: It should be indicated how many T cells were added to each tumoroid containing well.
- 2) Figure 7F, T cell recruitment by conditioned medium: It should be explained why it can be excluded that the observed recruiting effect stems from the bsAb remaining in the conditioned medium rather than from a chemotactic factor.
- 3) It would be nice to see a confirmation that change in FSC and SSC of T cells indicating T cell activation correlates with CD69 surface display which is a commonly used early marker for T cell activation

Reviewer #3

(Remarks to the Author)

The topic of the studies is comprehensive. The experiments are well-planned and the manuscript is organized effectively for readability.

The use of bispecific antibodies (bsAbs) generated through Duobody technology, with two arms - one binding to the TAA and another binding to CD3, represents a significant advancement in the therapeutic field. The novelty of the work lies in demonstrating the effects of bsAbs generated in breast cancer treatment, as well as in utilizing the fibrillar collagen-based ECM embedded tumoroid model for drug efficacy testing.

Utilizing the fibrillar collagen-based ECM embedded tumoroids model creates an environment resembling where breast cancers typically grow in the human body, which addresses the issue of inefficacy when bsAbs are used in patient treatments. Consequently, ECM embedded tumoroids serve as a valuable model for preliminary drug discovery studies before advancing to clinical trials. Moreover, this model is valuable and can be used to test drug efficacy prior to patient treatment selection.

Additionally, the use of confocal fluorescent microscopy and 3D image analysis in monitoring T cell recruitment and tumor killing clearly demonstrates that the efficacy of bsAbs is significantly influenced by CD3 affinity, the HER2 epitope used for the TAA arm as well as the concentration of bsAbs.

However, there are some recommended changes to a few of the sentences to improve clarity.

- Line 13: screening to screen
- Line 46: polyvinylidene fluoride
- Line 52: change the '>' to over
- Line 52: a comma (,) right after carcinomas to improve the readability.
- Line 58: change '~' to approximately
- Line 59: "and may also benefit from advanced therapies targeting Her2" requires some further clarification as the research suggests potential benefits from HER2-targeted therapies in HER2 low expressing breast cancers is not universally established yet. Thus, the statement may require a qualifier like "potentially" to accurately reflect the current understanding.
- Line 13, 16, 54, 55, 57, 59, 69: Terminology consistency. HER2 for the human epidermal growth factor receptor 2. Please check through the text and amend accordingly unless it is pointing to the her2 gene.
- Line 76: change 'prevent' to 'hinder'.
- Line 77: change 'E.g.' to 'For example'
- Line 97: change '~' to 'approximately'
- Provide reference for the statement "Mixed 3D co-cultures of tumor spheroids or organoids with T cells have advanced the field but do not allow monitoring of recruitment 321 of T cells from the environment towards the tumoroid." (pg 10, line 320-322)
- Justify the text from pg 11 onwards for consistency including the Figure and Table legend.

Author Rebuttal letter:

Rebuttal COMMSBIO-24-2462-T

We thank the editor and the reviewers for their time and effort to critically evaluate our manuscript. Please find below our response to the comments made by the reviewers. We believe these have been most helpful to further improve the quality of the paper.

In addition to the suggestions of the reviewers, we have considerably shortened the title and abstract as required and made other changes as requested in the "CommsBio-file-checklist-revision".

We include a version with all changes tracked and a clean version of the manuscript.

Reviewer #1

The authors address a novel important mechanism in tumor eradication by T-cell recruitment, using arrays of 3D tumoroids seeded in extracellular matrix. Here they screen for activity a series of bispecific antibodies of effector-silent Duobody format, which are directed against CD3 and Her2, and of variable affinities, and also directed against different epitopes of Her2. They discover that for certain “effective” bsAbs, the human healthy-donor derived T cells can be efficiently recruited after the initial tumor - T cell contact, which results in potent tumor killing. Their evidence is based on quantitative 3D-image analyses of confocal microscopy derived data. Interestingly, they observe not only efficient activation of T cells, but also proliferation after exposure to co-culture of target and T cells, which was not the case when the recruitment was elicited with CXCL12 cytokine.

These findings are highly important for future design of discovery of bispecific T-cell engaging antibodies, and the methodology presented here is very well supported by use of several controls, including low-expressing Her2 cell lines and knock-out control cell lines. Please find below a list of remarks which I hope you will find helpful.

1. Page 1: „T cell recruitment” – should be T-cell recruitment (please correct throughout the text). T cell is not hyphenated, but expressions such as T-cell recruitment, movement, etc. are hyphenated. We have corrected this as suggested.
2. Page 1: “despite equal high affinities” – equally high? Corrected as suggested.
3. Page 3: “immune checkpoint inhibitors can improve the response to CD3xHer2 bsAbs” – this statement is too general. Mentioning the opportunities given by involving immune checkpoints into therapeutic approaches with bispecifics might be beneficial. This was not meant as a general statement, but it is an actual description of the finding in the publication that is referred to at the end of this sentence.
4. Page 3: *in vitro/in vivo* should be in italics. Corrected as suggested.
5. Supplementary Figure 1: “1 collagen embedded BT474 tumoroid” – or MDA-MB231 tumoroid. Corrected as suggested.
6. Page 4: “tumor nuclei were detected (Hoechst) and subsequently scored as PI positive or negative” – I suppose the cells were scored as PI positive or negative. The sentence is correct: HOECHST was used to create a mask of the individual nuclei, which were then scored as PI pos or neg.
7. Page 4: In the figure 2C, the two shades of blue showing DII and DIII of Her2 are difficult to differentiate. This figure has been modified to enhance the contrast between DII and DIII as requested.
8. Page 4: Fc-inert, not Fc inert. At this point it should be explained which changes were made to obliterate the Fc function. Corrected as suggested. We now refer in this sentence to Table 1 where the Fc-inert mutations are specified in the legend.
9. Page 5: 1 ug/mL – micrograms should be correctly denoted, please correct throughout the manuscript. Corrected as suggested.
10. Page 5: CD3low: could the CD3 affinity for unmodified and low affinity variant be specified? We agree it is important to add this information. Affinities for CD3 WT and low affinity variant have been included in Table 1 and the legend to Table 1 is adjusted accordingly.
11. Page 11: “optimalization of bsAb efficacy is related to patient stratification” – in the further sentence, only Her2 expression on tumor is described, but it would be interesting to comment on the novel findings in view of immunologically cold tumors. Indeed, the data presented indicate potential chemotactic recruitment of T cells to “cold areas” provided that some initial engagement of a minor population of already present T cells by bsAbs can occur. We now extended this part of the revised discussion to comment on this aspect.
12. From page 12: all antibodies used should be listed with RRIDs. Added as suggested.
13. Page 15: Brilliant violet. Corrected as suggested.
14. Figure 3C. Western blots for Her2 and the control do not correspond to the same gels, or the discrepancies might be artefactual – please improve on the presentation of the results. The HER2 and Tubulin stainings shown were from the same Western blot. Perhaps the reviewer refers to the fact that the Tubulin blot is a bit overexposed and not perfectly aligned. In this revision, we now show a shorter Tubulin exposure (same experiment) and better horizontal alignment.
15. Byline to figure 6, page 29: MSD analysis of the total population of T cells in the vicinity of the tumoroid tumoroid over increasing time-lag from 1-30h – tumoroid only once. Corrected as suggested.
16. Page 32, legend to Table 1: I could not quite retrieve which of the variants from the reference 58 was used in this study. We refer here to data depicted in Table 1 of reference 59 (this was ref 58 in the first draft) where affinities of Herceptin and the bD2 variant (L in LbD2 stands for Light chain) are compared, amongst other variants.

The authors developed a platform, which allows one to generate 3D tumoroids embedded in collagen matrix to which T cells can be added for evaluation of T cell recruitment, activation and tumor cell killing. The authors show that different bispecific anti-Her2-CD3 antibodies display different abilities to kill tumor cells depending on their affinity to Her2 and CD3 and also depending on the location of the targeted Her2 epitopes. They could confirm previous findings that membrane-proximal epitopes elicit higher T cell activation, likely due to the formation of a tighter cytolytic synapse. The authors could trace the migration of single T cells to tumor cells and also provide evidence that chemotactic factor(s) play a role in T cell recruitment and accumulation at the tumoroid. Altogether, these are extremely interesting results and the presented platform allows for high throughput screening of bispecific antibodies for efficient tumor cell killing in a 3D tumor model, which is of great value for therapeutic antibody development. The paper is very well written and the experiments and their statistical analysis are sound and conclusive. The manuscript definitely merits publication.

Minor points:

1) Figure 3: It should be indicated how many T cells were added to each tumoroid containing well. This was mentioned in the Methods section under ECM embedded tumoroid arrays and exposure to T-cells and bsAbs: "For T-cell co-cultures, ~50,000 CMFDA-labeled T-cells were added on top of the collagen gels in presence or absence of bsAbs."

2) Figure 7F, T cell recruitment by conditioned medium: It should be explained why it can be excluded that the observed recruiting effect stems from the bsAb remaining in the conditioned medium rather than from a chemotactic factor. As no tumor cells are present in this setup to support the crosslinking of CD3:TAA bsAbs needed for activation of T-cells, their mere presence in the conditioned media would not trigger T cell recruitment. In this revision, we comment on this in the results section on page 9: For activation of T-cells, CD3 bsAbs require crosslinking through interaction with their other arm with the TAA on tumor cells. Hence, their presence in the conditioned media per se, would not trigger T cell recruitment.

3) It would be nice to see a confirmation that change in FSC and SSC of T cells indicating T cell activation correlates with CD69 surface display which is a commonly used early marker for T cell activation. We have performed the requested experiment, and the results are now shown in Supplementary Figure 2C. As anticipated by the reviewer, indeed T cell activation induced by coculture with tumor cells in the presence of bsAbs, leads to increased FSC/SSC as well as expression of CD69 as detected by FACS.

Reviewer #3 (Remarks to the Author):

The topic of the studies is comprehensive. The experiments are well-planned and the manuscript is organized effectively for readability.

The use of bispecific antibodies (bsAbs) generated through Duobody technology, with two arms - one binding to the TAA and another binding to CD3, represents a significant advancement in the therapeutic field. The novelty of the work lies in demonstrating the effects of bsAbs generated in breast cancer treatment, as well as in utilizing the fibrillar collagen-based ECM embedded tumoroid model for drug efficacy testing.

Utilizing the fibrillar collagen-based ECM embedded tumoroids model creates an environment resembling where breast cancers typically grow in the human body, which addresses the issue of inefficacy when bsAbs are used in patient treatments. Consequently, ECM embedded tumoroids serve as a valuable model for preliminary drug discovery studies before advancing to clinical trials. Moreover, this model is valuable and can be used to test drug efficacy prior to patient treatment selection.

Additionally, the use of confocal fluorescent microscopy and 3D image analysis in monitoring T cell recruitment and tumor killing clearly demonstrates that the efficacy of bsAbs is significantly influenced by CD3 affinity, the HER2 epitope used for the TAA arm as well as the concentration of bsAbs.

However, there are some recommended changes to a few of the sentences to improve clarity.

- Line 13: screening to screen. Corrected as suggested.

- Line 46: polyvinylidene fluoride. Corrected as suggested.

- Line 52: change the '>' to over. Corrected as suggested.

- Line 52: a comma (,) right after carcinomas to improve the readability. Corrected as suggested.

- Line 58: change '~' to approximately. Corrected as suggested.

- Line 59: "and may also benefit from advanced therapies targeting Her2" requires some further clarification as the research suggests potential benefits from HER2-targeted therapies in HER2 low expressing breast cancers is not universally established yet. Thus, the statement may require a qualifier like "potentially" to accurately reflect the current

understanding. Corrected as suggested.

- Line 13, 16, 54, 55, 57, 59, 69: Terminology consistency. HER2 for the human epidermal growth factor receptor 2. Please check through the text and amend accordingly unless it is pointing to the her2 gene. Corrected as suggested.

- Line 76: change 'prevent' to 'hinder'. Corrected as suggested.

- Line 77: change 'E.g.' to 'For example'. Corrected as suggested.

- Line 97: change '~' to 'approximately'. Corrected as suggested.

- Provide reference for the statement "Mixed 3D co-cultures of tumor spheroids or organoids with T cells have advanced the field but do not allow monitoring of recruitment 321 of T cells from the environment towards the tumoroid." (pg 10, line 320-322). The references have been added as suggested.

- Justify the text from pg 11 onwards for consistency including the Figure and Table legend. Corrected as suggested.

Version 1:

Reviewer comments:

Reviewer #1

(Remarks to the Author)

The authors have revised the manuscript according to the suggestions and improved minor issues in data presentation, as well as augmented the discussion, and with this addressed all my concerns.

Reviewer #2

(Remarks to the Author)

In the revised version, the authors addressed all critical point that were raised by the reviewers. It is nice to see that additional experiments were performed that further support the validity of their 3D tumoroid model.

Reviewer #3

(Remarks to the Author)

All the comments are well-addressed.

Author Rebuttal letter:

Rebuttal COMMSBIO-24-2462-T

We thank the editor and the reviewers for their time and effort to critically evaluate our manuscript. Please find below our response to the comments made by the reviewers. We believe these have been most helpful to further improve the quality of the paper.

In addition to the suggestions of the reviewers, we have considerably shorted the title and abstract as required and made other changes as requested in the "CommsBio-file-checklist-revision".

We include a version with all changes tracked and a clean version of the manuscript.

Reviewer #1

The authors address a novel important mechanism in tumor eradication by T-cell recruitment, using arrays of 3D tumoroids seeded in extracellular matrix. Here they screen for activity a series of bispecific antibodies of effector-silent Duobody format, which are directed against CD3 and Her2, and of variable affinities, and also directed against different epitopes of Her2. They discover that for certain "effective" bsAbs, the human healthy-donor derived T cells can be efficiently recruited after the initial tumor - T cell contact, which results in potent tumor killing. Their evidence is based on quantitative 3D-image analyses of confocal microscopy derived data. Interestingly, they observe not only efficient activation of T cells, but also proliferation after exposure to co-culture of target and T cells, which was not the case when the recruitment was elicited with CXCL12 cytokine.

These findings are highly important for future design of discovery of bispecific T-cell engaging antibodies, and the methodology presented here is very well supported by use of several controls, including low-expressing Her2 cell lines and knock-out control cell lines. Please find below a list of remarks which I hope you will find helpful.

1. Page 1: „T cell recruitment” – should be T-cell recruitment (please correct throughout the text). T cell is not hyphenated, but expressions such as T-cell recruitment, movement, etc.

are hyphenated. We have corrected this as suggested.

2. Page 1: "despite equal high affinities" – equally high? Corrected as suggested.

3. Page 3: "immune checkpoint inhibitors can improve the response to CD3xHer2 bsAbs" – this statement is too general. Mentioning the opportunities given by involving immune checkpoints into therapeutic approaches with bispecifics might be beneficial. This was not meant as a general statement, but it is an actual description of the finding in the publication that is referred to at the end of this sentence.

4. Page 3: *in vitro/in vivo* should be in italics. Corrected as suggested.

5. Supplementary Figure 1: "1 collagen embedded BT474 tumoroid" – or MDA-MB231 tumoroid. Corrected as suggested.

6. Page 4: "tumor nuclei were detected (Hoechst) and subsequently scored as PI positive or negative" – I suppose the cells were scored as PI positive or negative. The sentence is correct: HOECHST was used to create a mask of the individual nuclei, which were then scored as PI pos or neg.

7. Page 4: In the figure 2C, the two shades of blue showing DII and DIII of Her2 are difficult to differentiate. This figure has been modified to enhance the contrast between DII and DIII as requested.

8. Page 4: Fc-inert, not Fc inert. At this point it should be explained which changes were made to obliterate the Fc function. Corrected as suggested. We now refer in this sentence to Table 1 where the Fc-inert mutations are specified in the legend.

9. Page 5: 1 ug/mL – micrograms should be correctly denoted, please correct throughout the manuscript. Corrected as suggested.

10. Page 5: CD3^{low}: could the CD3 affinity for unmodified and low affinity variant be specified? We agree it is important to add this information. Affinities for CD3 WT and low affinity variant have been included in Table 1 and the legend to Table 1 is adjusted accordingly.

11. Page 11: "optimalization of bsAb efficacy is related to patient stratification" – in the further sentence, only Her2 expression on tumor is described, but it would be interesting to comment on the novel findings in view of immunologically cold tumors. Indeed, the data presented indicate potential chemotactic recruitment of T cells to "cold areas" provided that some initial engagement of a minor population of already present T cells by bsAbs can occur. We now extended this part of the revised discussion to comment on this aspect.

12. From page 12: all antibodies used should be listed with RRIDs. Added as suggested.

13. Page 15: Brilliant violet. Corrected as suggested.

14. Figure 3C. Western blots for Her2 and the control do not correspond to the same gels, or the discrepancies might be artefactual – please improve on the presentation of the results. The HER2 and Tubulin stainings shown were from the same Western blot. Perhaps the reviewer refers to the fact that the Tubulin blot is a bit overexposed and not perfectly aligned. In this revision, we now show a shorter Tubulin exposure (same experiment) and better horizontal alignment.

15. Byline to figure 6, page 29: MSD analysis of the total population of T cells in the vicinity of the tumoroid over increasing time-lag from 1-30h – tumoroid only once. Corrected as suggested.

16. Page 32, legend to Table 1: I could not quite retrieve which of the variants from the reference 58 was used in this study. We refer here to data depicted in Table 1 of reference 59 (this was ref 58 in the first draft) where affinities of Herceptin and the bD2 variant (L in LbD2 stands for Light chain) are compared, amongst other variants.

Reviewer #2 (Remarks to the Author):

The authors developed a platform, which allows one to generate 3D tumoroids embedded in collagen matrix to which T cells can be added for evaluation of T cell recruitment, activation and tumor cell killing. The authors show that different bispecific anti-Her2-CD3 antibodies display different abilities to kill tumor cells depending on their affinity to Her2 and CD3 and also depending on the location of the targeted Her2 epitopes. They could confirm previous findings that membrane-proximal epitopes elicit higher T cell activation, likely due to the formation of a tighter cytolytic synapse. The authors could trace the migration of single T cells to tumor cells and also provide evidence that chemotactic factor(s) play a role in T cell recruitment and accumulation at the tumoroid. Altogether, these are extremely interesting results and the presented platform allows for high throughput screening of bispecific antibodies for efficient tumor cell killing in a 3D tumor model, which is of great value for therapeutic antibody development. The paper is very well written and the experiments and their statistical analysis are sound and conclusive. The manuscript definitely merits publication.

Minor points:

1) Figure 3: It should be indicated how many T cells were added to each tumoroid containing well. This was mentioned in the Methods section under ECM embedded tumoroid arrays

and exposure to T-cells and bsAbs: "For T-cell co-cultures, ~50,000 CMFDA-labeled T-cells were added on top of the collagen gels in presence or absence of bsAbs."

2) Figure 7F, T cell recruitment by conditioned medium: It should be explained why it can be excluded that the observed recruiting effect stems from the bsAb remaining in the conditioned medium rather than from a chemotactic factor. As no tumor cells are present in this setup to support the crosslinking of CD3:TAA bsAbs needed for activation of T-cells, their mere presence in the conditioned media would not trigger T cell recruitment. In this revision, we comment on this in the results section on page 9: For activation of T-cells, CD3 bsAbs require crosslinking through interaction with their other arm with the TAA on tumor cells. Hence, their presence in the conditioned media per se, would not trigger T cell recruitment.

3) It would be nice to see a confirmation that change in FSC and SSC of T cells indicating T cell activation correlates with CD69 surface display which is a commonly used early marker for T cell activation. We have performed the requested experiment, and the results are now shown in Supplementary Figure 2C. As anticipated by the reviewer, indeed T cell activation induced by coculture with tumor cells in the presence of bsAbs, leads to increased FSC/SSC as well as expression of CD69 as detected by FACS.

Reviewer #3 (Remarks to the Author):

The topic of the studies is comprehensive. The experiments are well-planned and the manuscript is organized effectively for readability.

The use of bispecific antibodies (bsAbs) generated through Duobody technology, with two arms - one binding to the TAA and another binding to CD3, represents a significant advancement in the therapeutic field. The novelty of the work lies in demonstrating the effects of bsAbs generated in breast cancer treatment, as well as in utilizing the fibrillar collagen-based ECM embedded tumoroid model for drug efficacy testing.

Utilizing the fibrillar collagen-based ECM embedded tumoroids model creates an environment resembling where breast cancers typically grow in the human body, which addresses the issue of inefficacy when bsAbs are used in patient treatments. Consequently, ECM embedded tumoroids serve as a valuable model for preliminary drug discovery studies before advancing to clinical trials. Moreover, this model is valuable and can be used to test drug efficacy prior to patient treatment selection.

Additionally, the use of confocal fluorescent microscopy and 3D image analysis in monitoring T cell recruitment and tumor killing clearly demonstrates that the efficacy of bsAbs is significantly influenced by CD3 affinity, the HER2 epitope used for the TAA arm as well as the concentration of bsAbs.

However, there are some recommended changes to a few of the sentences to improve clarity.

- Line 13: screening to screen. Corrected as suggested.
- Line 46: polyvinylidene fluoride. Corrected as suggested.
- Line 52: change the '>' to over. Corrected as suggested.
- Line 52: a comma (,) right after carcinomas to improve the readability. Corrected as suggested.
- Line 58: change '~' to approximately. Corrected as suggested.
- Line 59: "and may also benefit from advanced therapies targeting Her2" requires some further clarification as the research suggests potential benefits from HER2-targeted therapies in HER2 low expressing breast cancers is not universally established yet. Thus, the statement may require a qualifier like "potentially" to accurately reflect the current understanding. Corrected as suggested.
- Line 13, 16, 54, 55, 57, 59, 69: Terminology consistency. HER2 for the human epidermal growth factor receptor 2. Please check through the text and amend accordingly unless it is pointing to the her2 gene. Corrected as suggested.
- Line 76: change 'prevent' to 'hinder'. Corrected as suggested.
- Line 77: change 'E.g.' to 'For example'. Corrected as suggested.
- Line 97: change '~' to 'approximately'. Corrected as suggested.
- Provide reference for the statement "Mixed 3D co-cultures of tumor spheroids or organoids with T cells have advanced the field but do not allow monitoring of recruitment of T cells from the environment towards the tumoroid." (pg 10, line 320-322). The references have been added as suggested.
- Justify the text from pg 11 onwards for consistency including the Figure and Table legend. Corrected as suggested.
